# Verification of Pre-Monsoon Temperature Forecasts over India during 2016 with focus on Heat Wave Prediction

Harvir Singh[1], Kopal Arora[2], Raghavendra Ashrit[1] and EN Rajagopal[1]

[1] National Centre for Medium Range Weather Forecasting, Ministry of Earth Sciences, Noida, 201309, India
5  [2]Swiss Re Global Business Solutions, Bangalore, 560027, India

*Correspondence to*: Harvir Singh (harviriitkgp@gmail.com)

**Abstract.** The operational medium-range weather forecasting based on Numerical Weather Prediction (NWP) models are complemented by the forecast products based on Ensemble Prediction Systems (EPS). This change has been recognized as an essentially useful tool for the medium range forecasting and is now finding its place in forecasting the extreme events. 10  Here we investigate extreme events (heat waves) using a high-resolution NWP models and its ensemble models in union with the classical statistical scores to serve the verification purposes. With the advent of climate change related studies in the recent past, the rising extreme events and their plausible socio-economic effects have encouraged the need for forecasting and verification of extremes. Applying the traditional verification scores and associated methods on both, the deterministic and the ensemble forecast, we attempted to examine performance of the ensemble based approach as compared to the 15  traditional deterministic method. The results indicate towards an appreciable competence of the ensemble forecasting detecting extreme events as compared to the deterministic forecast. Locations of the events are also better captured by the ensemble forecast. Further, it is found that the EPS smoothes down the unexpectedly soaring signals, which thereby reduce the false alarms and thus prove to be more reliable than the deterministic forecast.

## 1 Introduction

20  Reliable weather forecasting plays a pivotal role in our everyday activities. Over the years Numerical Weather Prediction (NWP) systems have been employed to serve the purpose. While the NWP models have demonstrated an improved forecasting capability in general, they still have a challenge in the accurate prediction of severe weather/extreme events. Severe weather events (thunderstorms, cloudburst, heat waves, and cold waves, etc.) usually involve strong non-linear interactions, often between small scale features in the atmosphere (Legg and Mylne, 2004 ). For example, development of 25  deep convection and thunderstorms in the tropics. These small-scale interactions are difficult to predict accurately (Meehl et al., 2001) and a small deviation in these could lead to completely different results, as a result of the forecast evolution process (Lorenz, 1969). The inherent uncertainty in the weather and climate forecasts can be well handled by employing ensemble based forecasting (Buizza et al., 2005). The Ensemble Prediction System (EPS) (Mureau et al., 1993; Molteni et al., 1996; Toth and Kalnay, 1997) were first introduced in the 1990's in an effort to quantify the uncertainty caused by the 30  synoptic scale baroclinic instabilities in the medium range weather forecasting (Legg and Mylne, 2004). Ensemble

forecasting has emerged as the practical way of estimating the forecast uncertainty and making the probabilistic forecasts (Buizza et al., 2005 and Ashrit et al., 2013). It is based on multiple perturbed initial conditions, ensemble approach samples the errors in the initial conditions to estimate the forecast uncertainty (spread in member forecasts). The skill of the ensemble forecast shows marked improvement over the deterministic forecast when comparing the ensemble mean to the deterministic forecast after a short lead time (Buizza et al., 2005).

The new EPS at the NCMRWF is now running for operational purposes since November 2015. This global medium-range weather forecasting system has been adopted from the UK Met Office (Sarkar et al., 2016). Generally, the model and the ensemble forecast applications in addition to their verifications are used for prevalent events with a limited focus on the rare extreme weather events. It would be for the first time that the EPS technique has been employed from this model output for the extreme events over India to study the heat wave events. The heat wave is considered if maximum temperature of a station reaches at least 40°C or more for Plains and at least 30°C or more for Hilly regions. Based on departure from normal, a station is declared to have heat wave conditions if departure from normal is 4.5°C to 6.4°C and severe heat wave if the departure from normal is >6.4°C. In terms of the actual maximum temperature, a station is under heat wave when actual maximum temperature ≥ 45°C and severe heat wave when the maximum temperature is >47°C. There has been increasing interest among the general public, media and local administration in predicting such extremes, the heat wave and cold wave events in India due to the associated loss of life. An increasing number of extreme temperature events over India were documented in several recent studies (Alexander et al., 2006; Kothawale et al., 2010; Hartmann et al., 2013; Rohini et al., 2016; Mehdi and Dehkale 2016) in a climatological study of the heat/cold wave show that over the Indian sub-continent between 1969 and 2013 there were more frequent cold and heat wave events over the Indo-Gangetic plains of India. In another study carried out for entire South Asia, Sheik et al., (2015) have reported that warm extremes have become more common and cold extremes less common.

The global temperatures have exhibited a warming trend of about 0.85°C due to anthropogenic activities between 1880 and 2012 (IPCC, 2013 and Rohini et al., 2016). Similar trends were also observed in India with the annual air surface temperature rises during the $20^{th}$ century. This is evident from the detailed study presented in Kothawale et al., (2010) based on the data from 1901-2007. The study (Kothawale et al., 2010) shows that Indian mean, maximum and minimum annual temperatures have significantly increased by 0.51, 0.71 and $0.27^{\circ}$C per 100 years respectively, during 1901-2007. However, an accelerated warming was observed during 1971-2007, mainly due to the decade 1998-2007. The study (Kothawale et al., 2010) highlights that the mean temperature during the pre-monsoon season (March-May) shows an increasing trend of $0.42^{\circ}$C per 100 years. On the other hand, a recently reiterated IPCC report (2013) notified an "unequivocal" proof of the increasing warming trend, globally which could be associated with the variations in the climate system. This indicates a need to comprehend the heat wave events on weather and climatic scales.

Current paper attempts to demonstrate the capability and strength of predicting such events using both ensemble and deterministic models. Present study investigates the heat wave events during the summer months March, April & May

(MAM) 2016 in India. This study considers two cases to demonstrate the strength and weaknesses of the EPS approach in predicting such extreme events.

With these factors in mind, we can say that temperature (Minimum and Maximum both), forms a vital component of weather and climatic studies which are becoming increasingly important and challenging. Reliable projections of such changes in our weather and climate are critical for adaption and mitigation planning by the agencies involved. The knowledge would undoubtedly be useful for a layman and the society. Forecast verification plays an important role in addressing two main questions: How good is a forecast? And how much confidence can we have in it?

Verification by employing statistical scores is a well-established and recommended method by World Meteorological Organization (WMO) adopted in this study (WMO 2008). This is the challenging situation when one needs to decide how much confidence can be placed in a model. Depending upon the statistical characteristics of the variable addressed, the score type is chosen and is employed for the verification. This fact offers a choice and challenge to adopt the most compatible score type. The set of verification scores used here are listed and briefly discussed in the next section.

In this paper, we investigate the utility of the ensemble prediction system over the deterministic forecast in studying extreme events like heat waves. This forms the first documented study of the 2016 heat wave events over India which was verified using the deterministic and the ensemble forecasts. This also provides some important insights into use of the ensemble forecast over the deterministic forecast in predicting extreme events like a heat wave. However, this study is unable to encompass an entire discussion on the efficiency of the EPS in general as the work examines a narrow range of phenomena over a not so wider region.

The paper begins with a brief explanation of the observed temperature (*Tmax* & *Tmin*) data sets, model description and the methodology used. It will then go on to the results' section which encompasses two case studies from the 2016 heat wave events in India, followed by the verification results and finally ending with the discussions and conclusions.

## 2 Observation, Model description and verification methodology

### 2.1 Observed Temperature (Maximum and Minimum)

IMD has developed a high resolution daily gridded temperature dataset at 0.5° x 0.5° resolution, which is available for the users (http*://www.imdpune.gov.in/Seasons/Temperature/max/Max_Download.html*). Data processing procedure has been well documented (Srivastava et al., 2009). IMD has compiled, digitized, quality controlled and archived these data at the National Data Centre (NDC). Based on maximum data availability, some stations were subjected to quality control checks like rejecting values, greater than exceeding known extreme values, minimum temperature greater than maximum temperature, same temperature values for many consecutive days, etc. After these quality checks, 395 stations were selected for further development of gridded data. IMD used measurements at these selected stations and interpolated the data into grids with the modified version of Shepard's angular distance weighting algorithm (Shepard, 1968). In this study, we have used IMD's real-time daily gridded (Shepard, 1968; Piper and Stewart, 1996; New et al., 2000; Kiktev et al., 2003; Rajeevan

et al., 2005; Caesar et al., 2006 and Srivastava et al., 2009) temperature (maximum and minimum) data to verify the real-time forecasts based on NCMRWF Unified Model (NCUM; deterministic) and NCMRWF Ensemble Prediction System (NEPS) ensemble mean forecast temperatures. The verification is carried out for the entire period from March 2016 to May 2016 at $0.5^{o} \times 0.5^{o}$ resolution over Indian land area.

## 2.2 NCMRWF Unified Model (NCUM)

The Unified Model (John et al., 2016), operational at NCMRWF consists of an Observation processing system (OPS 30.1), four-dimensional variational data assimilation (VAR 30.1) and Unified Model (UM 8.5). This analysis system makes use of various conventional and satellite observations. The analysis produced by this data assimilation system is being used as initial condition for the daily operational high resolution (N768L70) global NCUM 10-day forecast since January 2016. The horizontal resolution of NCUM system is 17 km and has 70 levels in the vertical extends from surface to 80 km height. The NCUM model forecast temperature (*Tmax* & *Tmin*) data have been interpolated to the $0.5^{o} \times 0.5^{o}$ resolution using bilinear interpolation method to match the resolution and grids of the observed data.

## 2.3 NCMRWF Ensemble Prediction System (NEPS)

NEPS is a global medium-range ensemble forecasting system adapted from the UK Met Office MOGREPS system (Bowler et al., 2008). The configuration consists of four cycles of assimilation corresponding to 00Z, 06Z, 12Z & 18Z and 10-day forecasts are made using the 00Z initial condition. The N400L70 forecast model consists of 800x600 grid points on the horizontal surface and has 70 vertical levels. Horizontal resolution of the model is approximately 33 km in the mid-latitudes. The 10-day control forecast run starts with the operational NCUM (N768L70) analysis and 44 ensemble members start from different perturbed initial conditions consistent with the uncertainty in initial conditions. The initial perturbations are generated using Ensemble Transform Kalman Filter (ETKF) method (Bishop et al., 2001). Uncertainty in the forecasting model is taken into account by making small random variations to the model and using a stochastic kinetic energy backscatter scheme (Tennant et al., 2010).

## 2.4 Verification Metrics

There are several scores available for the categorical verification of forecasts (Wilks 2011). However, in the current study, we have used the Probability of Detection (POD), False Alarm Ratio (FAR), Equitable Threat Score (ETS), Hanssen and Kuipers Score (HK Score), and Symmertical Extremal Dependence Index (SEDI). A brief description of these scores is presented here.

### 2.4.1 Probability of Detection (POD) also known as Hit Rate (H)

The POD tries to answer the question, "*What fraction of the observed "yes" events were correctly forecasted?"* It is very much sensitive to hits, but ignores false alarms and very sensitive to the climatologically frequency of the event. It is good

for rare events and can be artificially improved by issuing more "yes" forecasts to increase the number of hits. Its value varies from 0 to 1, for perfectly forecasted events POD=1 and computed by the following Eq. (1):

$$POD = \frac{hits}{hits + misses},$$ (1)

### 2.4.2 False Alarm Ratio (FAR) (F)

*What fraction of the predicted "yes" events actually did not occur?* FAR is sensitive to false alarms, but ignores misses, very sensitive to the climatological frequency of the event and should be used in conjunction with the probability of detection. FAR computed by the Eq. (2):

$$FAR = \frac{hits}{hits - false\ alarms},$$ (2)

### 2.4.3 Hanssen and Kuipers Score (HK)

It reveals the true skill statistic and focuses on how well the forecast separates the "Yes" events from the "No" events. HK uses all elements in the contingency table, does not depend on climatological event frequency. The score ranges between -1 to 1, both inclusive along with 0, which indicates no skill and 1 denotes a perfect skill and computed by the Eq. (3):

$$HK = \left[ \frac{hits}{hits + misses} \right] - \left[ \frac{false\ alarms}{false\ alarms + correct\ negatives} \right],$$ (3)

This score is efficient at verifying the most frequent events. Temperature possesses continuous values just like precipitation
amount and a few other NWP variables. In such cases mean error, mean square error (MSE), root mean square error (RMSE), correlation and anomaly correlation are best suitable (4[th] international verification methods workshop, Helsinki, June 2009). Categorical values for instance precipitation occurrences are well suited for the verification analysis using POD, FAR, Heidke skill score, equitable threat score, and HK Score statistics. However, in order to take advantage of these scores, for our continuous variable, temperature (Maximum and Minimum), we categorize it using the temperature ranges,
30-32, 32-34, 34-36, 36-38, 38-40 and 40-42 °C.

### 2.4.4 Equitable Threat Score (ETS)

It is also known as the Gilbert skill score describe how well the forecasted "yes" events agree with the observed "Yes" events and thus exploring the hits by chance. This score ranges between -1/3 to 1. '0' shows no skill and 1 denotes the perfect skill. The score expresses the fraction of observed or the forecasted events projected accurately and ETS is computed by the
25 following Eq. (4):

$$ETS = \frac{hits - hits_{random}}{hits + false\ alarms - hits_{random}},$$ (4)

Where $hits_{random} = \frac{(hits - misse)(hits + false\ alarms)}{total}$

### 2.4.5 Symmetric Extremal Dependence Index (SEDI)

It expresses the association between a forecast and the observed rare events. It ranges between -1 and 1 where the perfect score is 1. This score converges to (2X -1) as the event frequency advance towards 0, where "X" denotes the variable that specifies the hit rate's convergence to 0 for the rarer events. SEDI is not influenced by the base rate SEDI score approaches 1 and computed by the following Eq. (5):

$$SEDI = \frac{\ln F - \ln H + \ln(1-H) - \ln(1-F)}{\ln F + \ln H + \ln(1-H) + \ln(1-F)},$$ (5)

Here H is Hit Rate (POD) and F is FAR.

## 3 Results and Discussions

The verification of temperature forecasts is presented in this section. The models are running operationally and are providing the forecasts out to 10 days every day. The verification is confined to MAM 2016, over six different temperature thresholds. For *Tmax*, the temperature thresholds are 32, 34, 36, 38, 40 & 42°C and for the *Tmin*, however, it is 22, 24, 26, 28, 30 & 32°C. The panels in Fig. 1a, b show the observed and forecast (day 3) frequency distribution for *Tmax* and *Tmin*. For lower temperature thresholds, the forecast underestimate the frequency, while, both, deterministic and ensemble mean converge towards observed relative frequency, especially for the temperature exceeding 38°C. The NEPS performs better than the NCUM forecast (Fig. 1a), indicating better performance of the ensemble model.

From the spatial map Fig. 2, the frequency of the observed maximum temperature *Tmax* ≥ 40°C over the Maharashtra and adjoining regions show maximum (more than 70 counts) over the entire period of MAM 2016, which is picked up by both deterministic and ensemble forecasts. However, the deterministic forecast is showing more frequency spread over MP, UP and Bihar, Odisha, AP and adjoining states from day 1 to day 9. As forecast lead time increases from day 1 to day 9 the heat wave frequency increases from central India to the north and east India. Consequently, a higher number of heat wave extremes was predicted by NCUM over east UP, Bihar, West-Bengal, Odisha, Jharkhand, Chhattisgarh, and AP. On the other hand, NEPS (Fig. 3) prediction for the day 1 to day 9 is much subdued than in the NCUM forecasts. However, both models, NCUM, and NEPS are, predicting more frequently the heat waves over the above-said regions. Comparatively, the ensemble-based model NEPS is performing better (spatially) for the extremes of heat wave events than the deterministic-based model NCUM over most of the Indian states up to day 9.

## 4 Case Studies for Extreme Heat waves

### 4.1 Weather conditions during March-May 2016

The heat wave conditions prevailed at some places over the central and adjoining western parts of the country during last week of March 2016 (Climate Diagnostics Bulletin of India, March 2016) and over parts of central and northwest India during the first week of April (Climate Diagnostics Bulletin of India, April 2016). These conditions prevailed over most

parts of east India all through the second week. According to IMD official reports the severity and extent of heating increased during the next week resulting in the establishment of severe heat wave conditions over parts of north and eastern India. These conditions continued to prevail over east India and also spread over parts of south India during the fourth week, however, its intensity and areal extent reduced towards the end of the week. During the last few days of the April month, heat wave conditions prevailed over parts of Odisha, Bihar, Gangetic West Bengal and Kerala. During the month of May 2016, heat wave conditions were reported at isolated places on some occasions over parts of Rajasthan, Punjab, Odisha, Gangetic West Bengal and Kerala during the first fortnight of the month (Climate Diagnostics Bulletin of India, May 2016). As per the May Climate Diagnostics Bulletin of India, severe heat wave conditions developed and intensified over parts of northwest India. From 15$^{th}$ May the heat wave conditions spread and persisted over parts of central and north peninsular India till 22$^{nd}$ of the month. Jammu & Kashmir, west & east Rajasthan, west & east Madhya Pradesh and Vidarbha were especially affected during this period. Some stations of west Rajasthan viz. Barmer, Bikaner, Ganganagar, Jaisalmer, and Jodhpur observed severe heat wave conditions for 4 to 5 days in succession from 17 to 21 May and temperature observed ≥ 50°C. The heat wave conditions gradually abated from most parts of the country after 23$^{rd}$ and prevailed only at isolated places over parts of Coastal AP and Telangana during last few days of the month.

## 4.2 Casualties reported during March-May 2016

As per the official IMD reports (Climate Diagnostics Bulletin of India, March, April and May 2016) the prevailing heat wave over India caused more than 500 loss of lives. Heat wave claimed one life each (Climate Diagnostics Bulletin of India, March 2016) in Maharashtra (Nanded, 13 March) & Kerala (Palakkad, 5 March). A brief account of heat wave-related deaths is listed in Table 2. It took a toll of over 200 lives (Climate Diagnostics Bulletin of India, April 2016) from central and peninsular India during the April month. Of these, 88 lives were reported from Odisha, 79 from Telangana, 40 from AP, 9 from Maharashtra and one each from Karnataka and Tamil Nadu. In the month of May heat wave claimed over 275 lives from central and peninsular parts of the country. Of these, over 200 lives (Climate Diagnostics Bulletin of India, April 2016) were reported from Telangana alone. 39 lives were reported from Gujarat and 34 from Maharashtra.

## 4.3 Synoptic features associated with Heat waves during 2016

The panels in Fig. 4 on the left show analysis (top) and day 3 forecast (bottom) MSLP and winds at 700 hPa for 10$^{th}$ April 2016. Similarly, the panels on the right show analysis (top) and day 3 forecast (bottom) MSLP and winds at 700 hPa for 21$^{st}$ May 2016. The typical synoptic features associated with the pre-monsoon season is depicted in the above figure, which shows the MSLP in hPa (shaded) and 700 hPa winds in ms$^{-1}$ (vectors) over Indian sub-continent. The low pressure associated with continental heating (*heat low*) is prominent and an important semi-permanent system that drives the monsoon (Rao, 1976). The heat low establishes over NW parts of India and adjoining Pakistan and is seen to extend over India. The day 1 and day 3 forecasts successfully capture this broad scale feature of the heat low. The 700 hPa winds over central India are predominantly north-westerlies driving the hot and dry air from over the Thar desert towards the central

India. The pre-monsoon hot weather gets severe at times when the hot and dry north-westerlies penetrate deep into the peninsula and persist for several days. During May 2016, similar conditions caused severe heat wave over parts of Maharastra, Telangana and Odisha.

### 4.3.1 Case-I Heat waves on 11th April 2016

As per the IMD reports (Climate Diagnostics Bulletin of India, April 2016), the heat wave conditions prevailed over parts of central peninsular and east India during the second week of the April. It took a toll of over 200 lives (Table-2) from central and peninsular India during the April month. Observed and forecast *Tmax* valid for 11th April 2016 is shown for NCUM (Fig. 5) and NEPS (Fig. 6). The spatial distributions of *Tmax* show prevailing heat waves over Odisha, AP, Telangana, and some parts of Maharashtra on 11th April 2016. The observation shows more than 40°C spread over east UP, Bihar, West Bengal, east MP, Jharkhand, Chhattisgarh, Odisha, Maharashtra, some parts of Karnataka and Tamil Nadu. In the NCUM forecast, on other hand showing marginally wider regions up to day 9 due to a warm bias in the model and on the contrary NEPS forecasts also are showing ≥40°C wider regions up to day 9 but marginally less than the NCUM forecasts. Apart from the warm bias, both the model forecasts are showing cold bias in north-northeast of J&K. Hence the NEPS is better in predicting the extremes of heat waves up to day 9 then the NCUM.

### 4.3.2 Case-II Heat waves on 21st May 2016

The severe heat wave conditions developed and intensified over parts of northwest India during third week of May 2016 and persisted over parts of central and north peninsular India some stations of west Rajasthan temperature observed ≥ 50°C viz. Barmer, Bikaner, Ganganagar, Jaisalmer & Jodhpur, observed the severe heat wave conditions for 4 to 5 days in succession from 17th May to 21st May in 2016. The spatial distributions of NCUM & NEPS forecast *Tmax* with observed IMD *Tmax,* prevailing heat waves over Rajasthan, MP, UP, Delhi, Haryana, Punjab and some parts of Maharashtra on 21st May 2016 is shown in Figs. 7 & 8. Both the models, deterministic and ensemble are able to predict the extreme temperature (*Tmax* > 48°C) over west Rajasthan up day 3 only. However, the NCUM is predicting more wide-spread *Tmax* > 46°C, over Rajasthan, MP, UP, Delhi, Haryana, Punjab and parts of Maharashtra all days forecast.

The HK scores of the maximum temperature (*Tmax*) between the range 30-42 °C, constructed as box and whiskers for both NCUM and NEPS, indicate towards better performance of the ensemble based forecast as compared to the deterministic one. Interestingly, the forecast score does not fade away with the lead time contrary to the expectation. This depicts that the NEPS performs better and its prediction skill remains quasi-constant throughout the lead time of 10 days (Fig. 9).

Similar observations can be made from the ETS plots (Fig. 10). The most obvious finding to emerge from the box and whiskers plots of the ETS scores is the better performance of the ensemble based forecast (NEPS) than that of the deterministic forecast (NCUM). This result is consistent with the earlier documented findings. At all the *Tmax* thresholds (between 30 and 42°C), NEPS mean stands above the NCUM mean. The same observation holds during all the illustrated

forecasts (day 1, 3, 5, 7, and 9). The scores falling under the 25% in the case of the ensemble based forecast are either similar or lie little above the deterministic forecast unlike the values underlying 75% which in the NEPS case are markedly higher than that of the NCUM's.

This finding raises an intriguing question regarding the difference in the characteristic distribution of both NEPS and NCUM forecasts. This result also advocates better performance of the ensemble based forecast over the deterministic forecast. Importantly, the ensemble-based forecast predicts low false alarm than its counterpart, NCUM, especially in the high-temperature range. In the low-temperature range, between 30 and 32, NEPS has low FAR score (where 0 denotes the perfect score) for day 1 and day 3 forecast. Similarly, a comparatively higher score on day 5, 7 and 9 respectively (Fig. 11).

Probability of detection of ensemble based forecast is higher than the deterministic forecast during all the lead times and at all the temperature thresholds except for the day 1 forecast score for the NEPS in the range between 40-42°C where NCUM shows better performance (Fig. 12).

 At higher temperature ranges, representing rare events, the performance of NEPS and NCUM can be clearly seen from the SEDI score plot (Fig. 13). We can notice a considerable difference between the performance of the two techniques for extreme events lying between 40 and 42 °C, on all the days.

Apparently, NEPS demonstrates higher skill than that of NCUM in predicting the heat wave events. The heat wave event prediction skill is best seen on the day 5 forecast with NEPS's SEDI score encompassing the score value of 0.7. Monthly scores also are listed in table 3.

A consistent result attained from the NEPS and NCUM verification demonstrates the better skill of the ensemble forecasts compared to the deterministic forecast for the considered cases.

## 5 Summary and Conclusions

Unless the atmosphere is in a highly predictable state, we should not expect an ensemble to forecast extreme events with a high probability (Legg and Mylne, 2004). This is due to the small scale non-linear interactions involved in a model (NWP). One of the several predicted interactions could be climatologically extreme and are hence more difficult to predict. A small variation in the intensity, timing and position of such anomalies could lead to a large difference in their prediction growth in time. Thus, despite the efficiency of EPS over the deterministic forecast in detecting extreme events, we should be extremely careful in declaring it locally as explained above. The ensemble mean is relatively better in predicting the extremes of heat wave events than the deterministic forecast overall Indian states up to day 9.

1) The ensemble forecast provides appreciable forecasts on all the days and is most reliable after the day 2 forecast. This characteristic is more pronounced for extreme events than for the less extreme events where the ensemble forecast after day 2 is less reliable as can be seen from the FAR and POD scores at the lower thresholds. This suggests that the performance of EPS on different thresholds is different that is, if it performs well at higher thresholds, it does not necessarily mean that it would perform equally well at the lower thresholds too. Thus, we need to understand the model

performance at all the concerned ranges and based upon those verification results, employ the ensemble forecast accordingly for operational purposes.

2) Our forecasts were obtained for the 2016 pre-monsoon season in India, MAM and since the severe events are rare in nature it limits the sample size for the ensemble forecast and thus poses a challenge for the efficient forecast verification. Despite the caveats involved, the ensemble forecast has shown to predict the heat waves several days ahead of the event, as discussed in the results. The severe heat waves (>40°C) can reliably be predicted for day 2 onwards with less false alarms as compared to the deterministic forecast as observed here. This could be explained by the inherent smoothing characteristic of the ensemble based prediction contrary to the deterministic one which in our case shows warm bias.

3) Comparatively, low efficiency of the ensemble based prediction on a shorter time scales (<day 2) propose that the ensemble prediction may need a longer duration of time for the perturbation growth. However, over a medium range forecast and for the extreme events like heat waves, the ensemble-based approach proves to be one of the most economic and effective tools.

For the present study, the data from the two models is available only from 2016. Ensemble based forecasts in real-time using the NEPS started in November 2015 at NCMRWF. For a robust and conclusive result, it is necessary that the study is based on the higher number of cases. This will be carried out in future.

The temperature data from the station's distribution are discussed in this paper which is used to obtain the gridded *Tmax* and *Tmin* data. It is indeed likely that some of the station extremes are smoothed out in the gridded data. It should also be noted that the station's data network is sparse 395 and often there are missing values. Gridded data field provides a continuous and gap-free data to work with.

Extreme events like heat waves are rare in nature and here we provided a general view of the two particular heat wave events (11 April & 21 May). From our experience as well as the forecast for the post heat wave event days, we can state that the skill of predicting an event with the initial conditions of no indication of severity is comparatively lower than when the signature is present in the initial conditions. Even before the event, there is some signature of it as can be seen in the Figs. (5, 6, 7 & 8). The overall prediction of warm conditions is nicely predicted but at closer lead times, the events are better predicted. Same can be seen in the box and whisker plots for ETS (and rest of the score plots as well). For instance, the skill of NEPS does not fall drastically from day 2 to day 7 and thus depicts a reasonable skill. So, overall the NEPS specifically, has a good skill in predicting the extreme event and is relatively robust.

**Acknowledgments**

The authors are grateful to India Meteorological Department (IMD) for providing the gridded observed *Tmax* and *Tmin* data in real-time, which is a very useful product for verification of NWP forecasts. The authors are thankful for discussions and feedback provided by colleagues at NCMRWF, which has helped in improving the quality of the manuscript. Thanks are due for anonymous reviewers for their comments and suggestions which have helped in revising the manuscript.

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

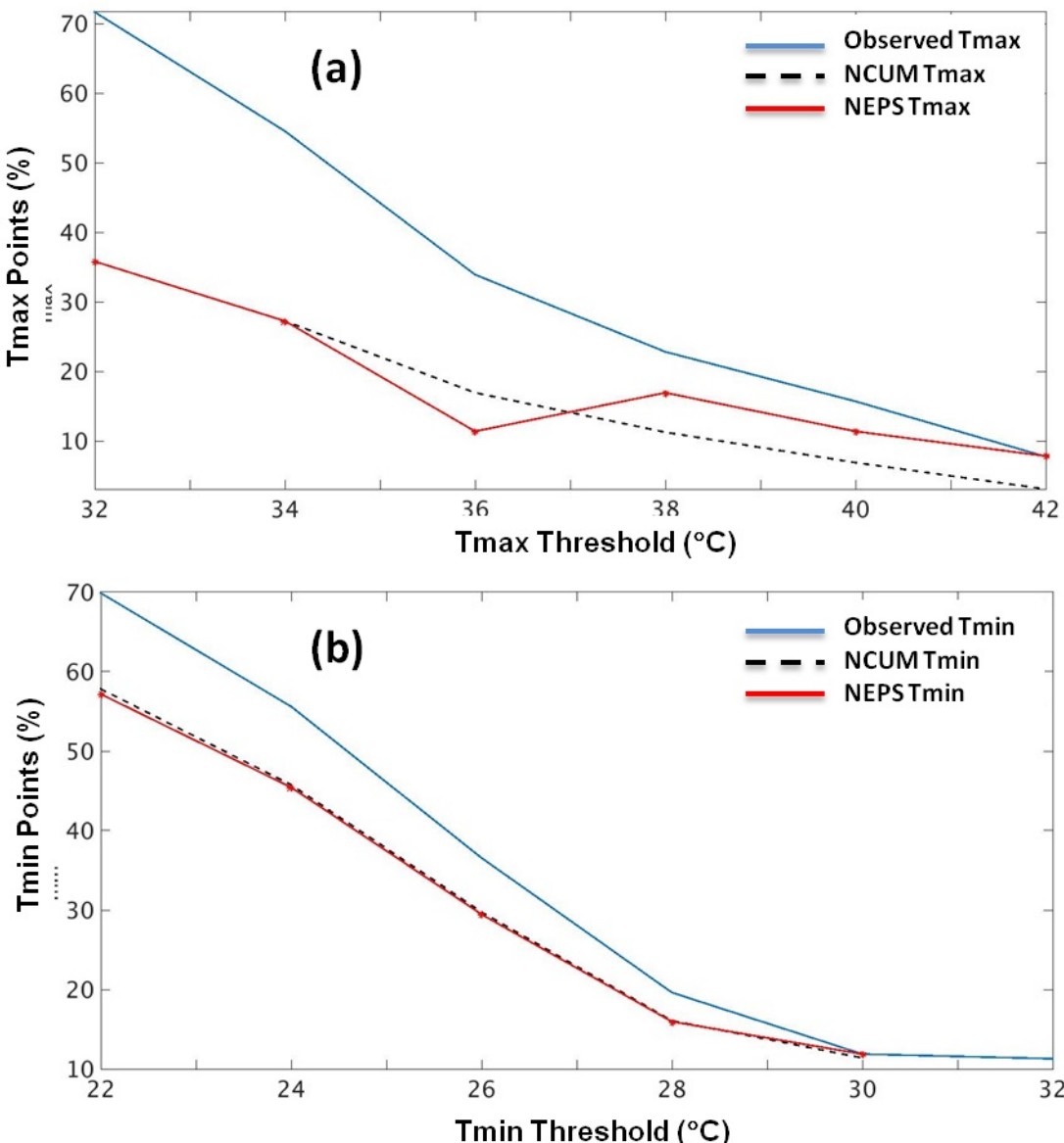

Figure 1: Frequency distribution of observed and forecast (NCUM and NEPS) (a) *Tmax* (°C) and (b) *Tmin* (°C) over India during March to May 2016. (NCUM stands for NCMRWF Unified Model and NEPS stands for NCMRWF Ensemble Prediction System) [Observed data from India Meteorological Department and forecast data from NCMRWF]. Both panels
5   indicate that the forecasts have lower frequency compared to observations at lower thresholds. At higher temperature thresholds (>40°C for *Tmax* and >28°C for *Tmin*), there is a better agreement.

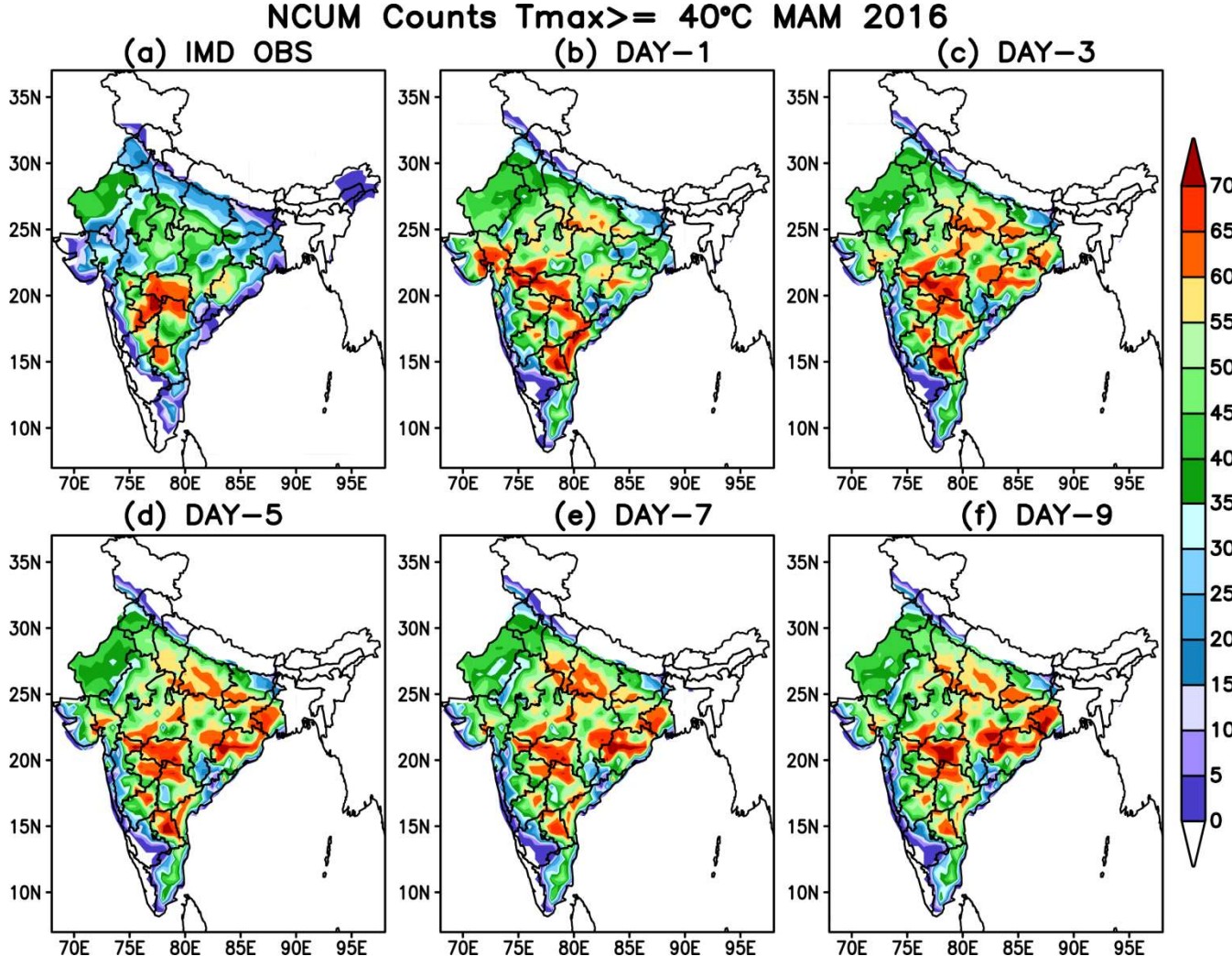

Figure 2: Spatial distribution of observed (a) and NCUM forecasts (b-f) number of days with *Tmax* ≥ 40°C during the period of March to May 2016. (NCUM stand for NCMRWF Unified Model) [Observed data from India Meteorological Department and forecast data from NCMRWF]. Regions with over 60 days of *Tmax* ≥ 40°C are indicated in orange and red shade. In the observations (Fig. 2a) this is confined to small part over peninsula. The NCUM forecasts (Fig. 2b-f) show *Tmax* ≥ 40°C widespread over northern and eastern parts of India.

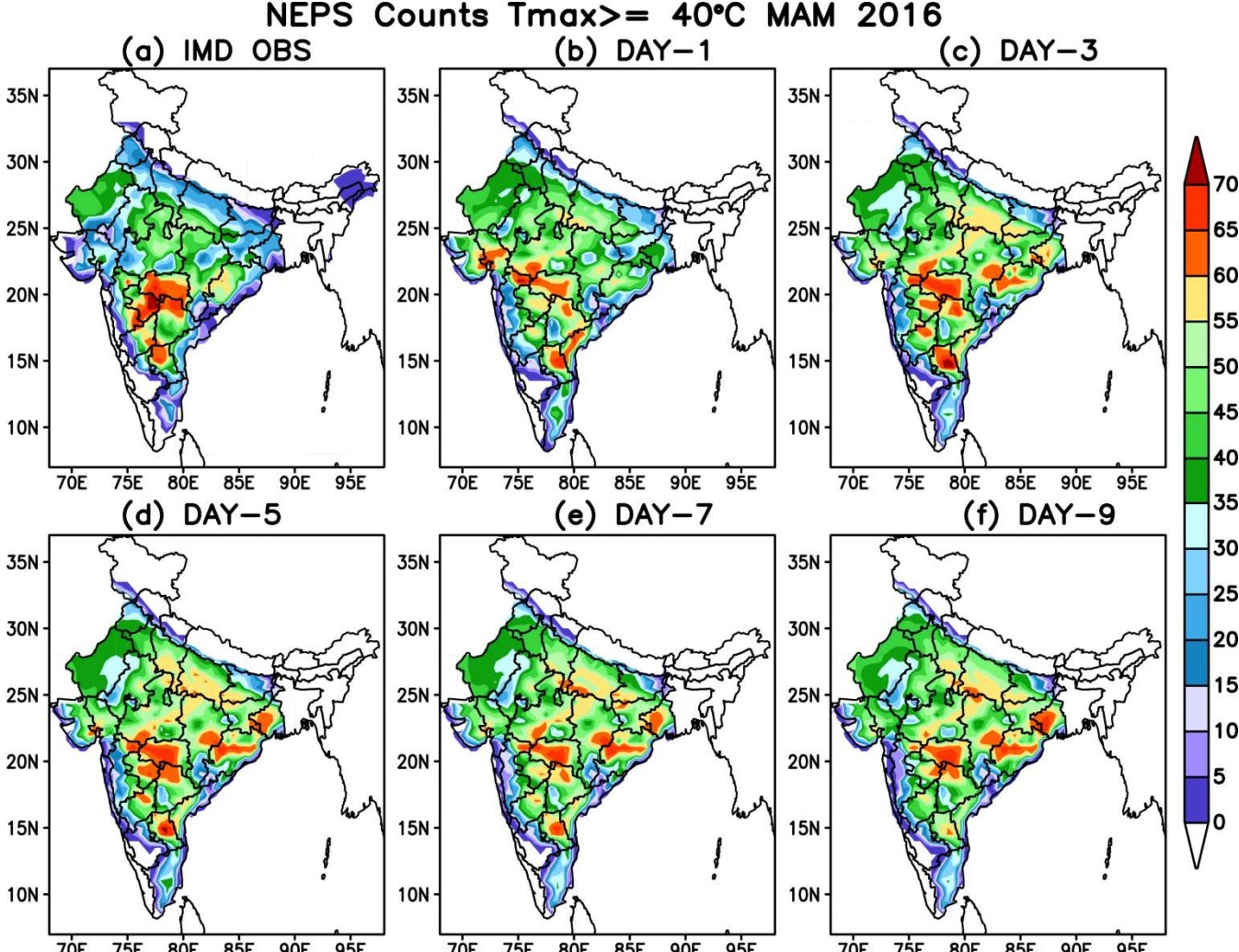

Figure 3: Spatial distribution of observed (a) and NEPS forecasts (b-f) number of days with *Tmax* ≥ 40°C during the period of March to May 2016. (NEPS stands for NCMRWF Ensemble Prediction System) [Observed data from India Meteorological Department and forecast data from NCMRWF]. Regions with over 60 days of *Tmax* ≥ 40°C are indicated in orange and red shade. In the observations (Fig. 3a) this is confined to small part over peninsula. The NEPS forecasts (Fig. 3b-f) show *Tmax* ≥ 40°C widespread over northern and eastern parts of India. NEPS forecasts (Fig. 3b-f) have a better agreement with observations (Fig. 3a) compared to NCUM forecasts (Fig. 2b-f).

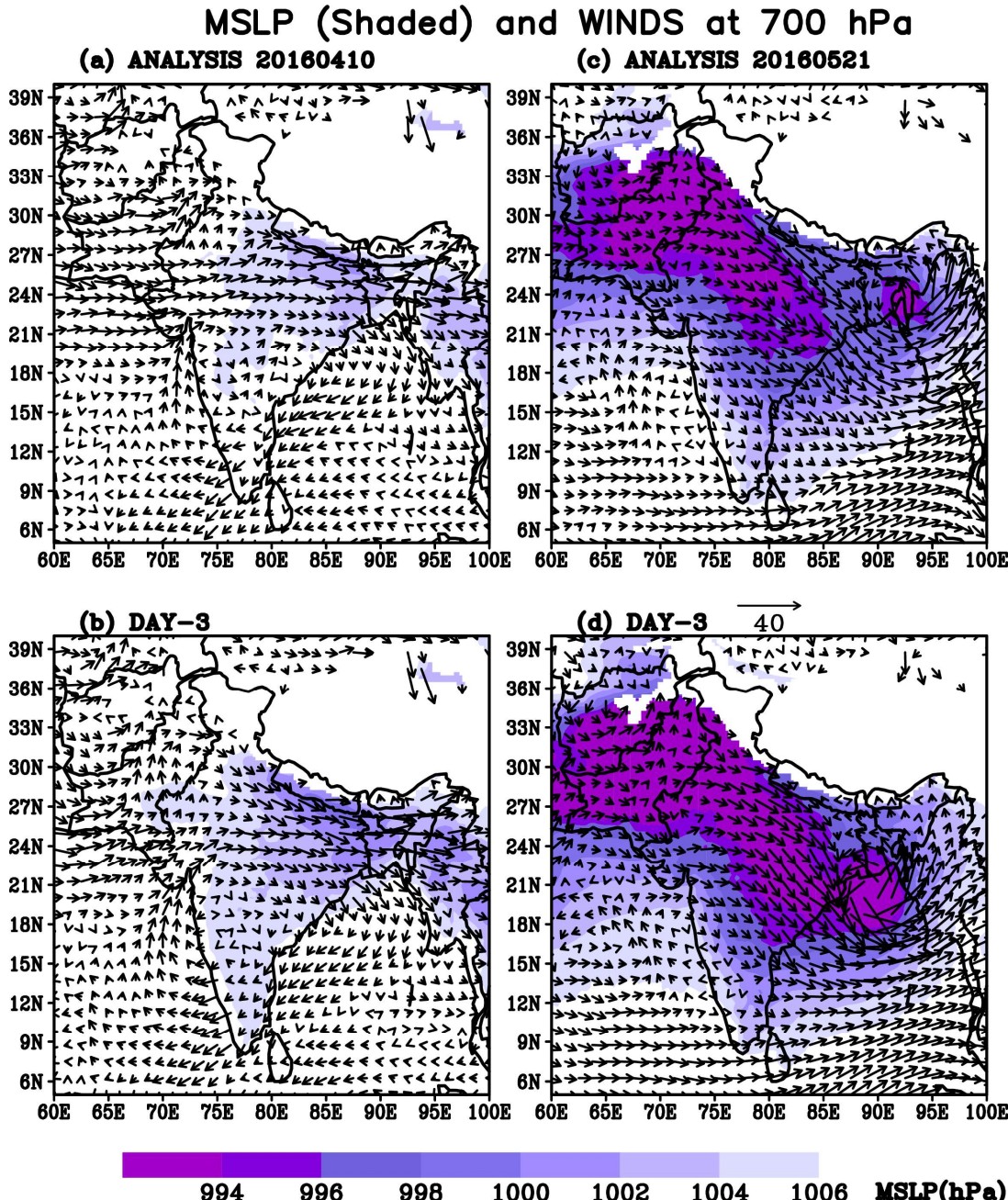

Figure 4: Mean Sea Level Pressure (MSLP) shaded and winds at 700 hPa showing heat low in the (a) Analysis of 20160410 (b) day 3 forecast valid for 20160410 (c) Analysis of 20160521 (d) day 3 forecast valid for 20160521. The analysis and the forecasts are based on the NCMRWF Unified Model. Both analysis (Fig. 4a,b) and forecasts (Fig. 4c,d) feature dry north-westerlies.

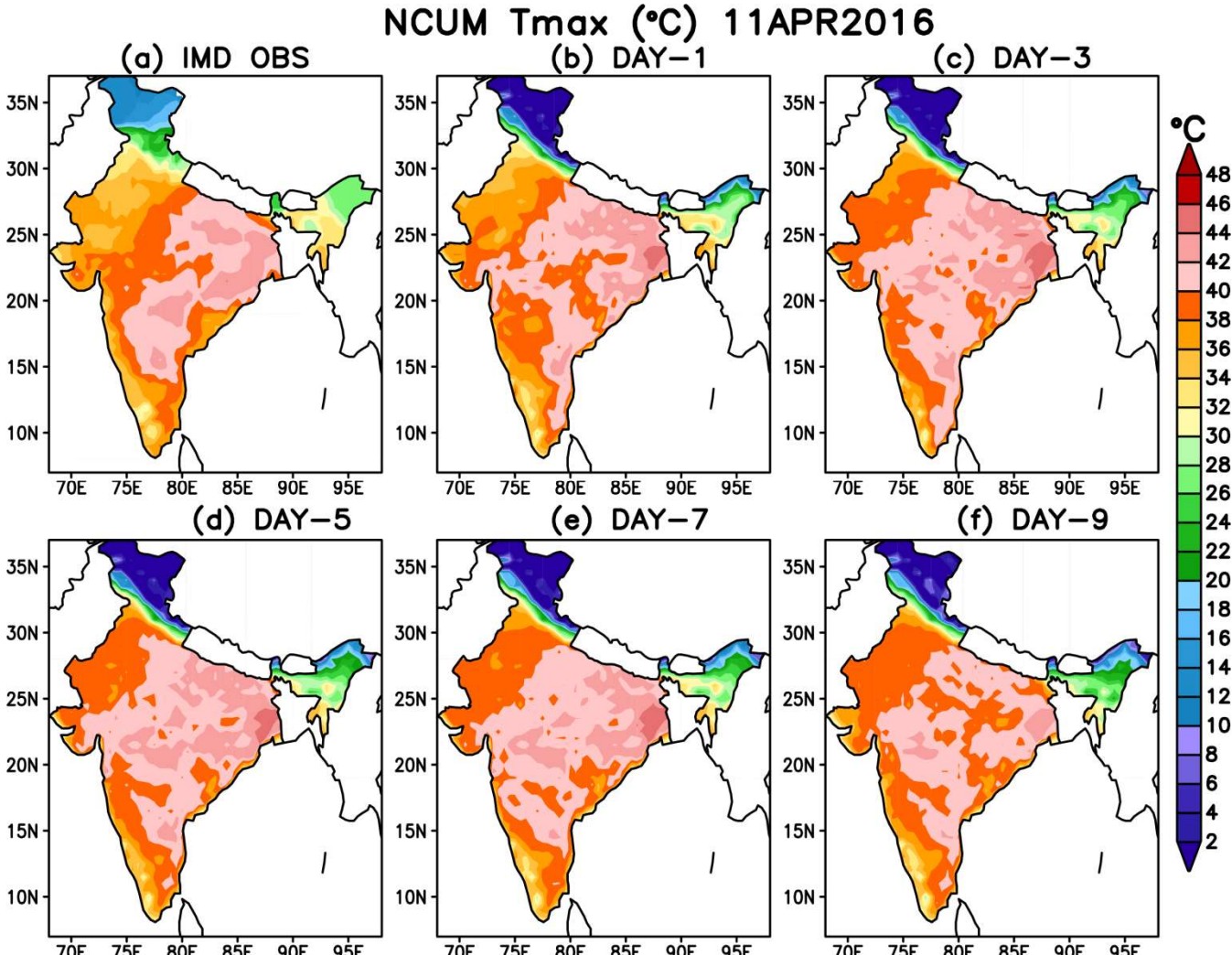

Figure 5: Spatial distributions of Observed *Tmax* (a) and NCUM forecast *Tmax* (b-f) prevailing heat waves over, MP, Odisha, AP, Telangana and some parts of Maharashtra on 11[th] April 2016. (NCUM stand for NCMRWF Unified Model). [Observed data from India Meteorological Department and forecast data from NCMRWF]. Observed (Fig. 5a) *Tmax* ≥ 40°C is widespread over eastern India and large part of peninsula. NCUM forecasts (Fig. 5b-f) also show similar distribution.

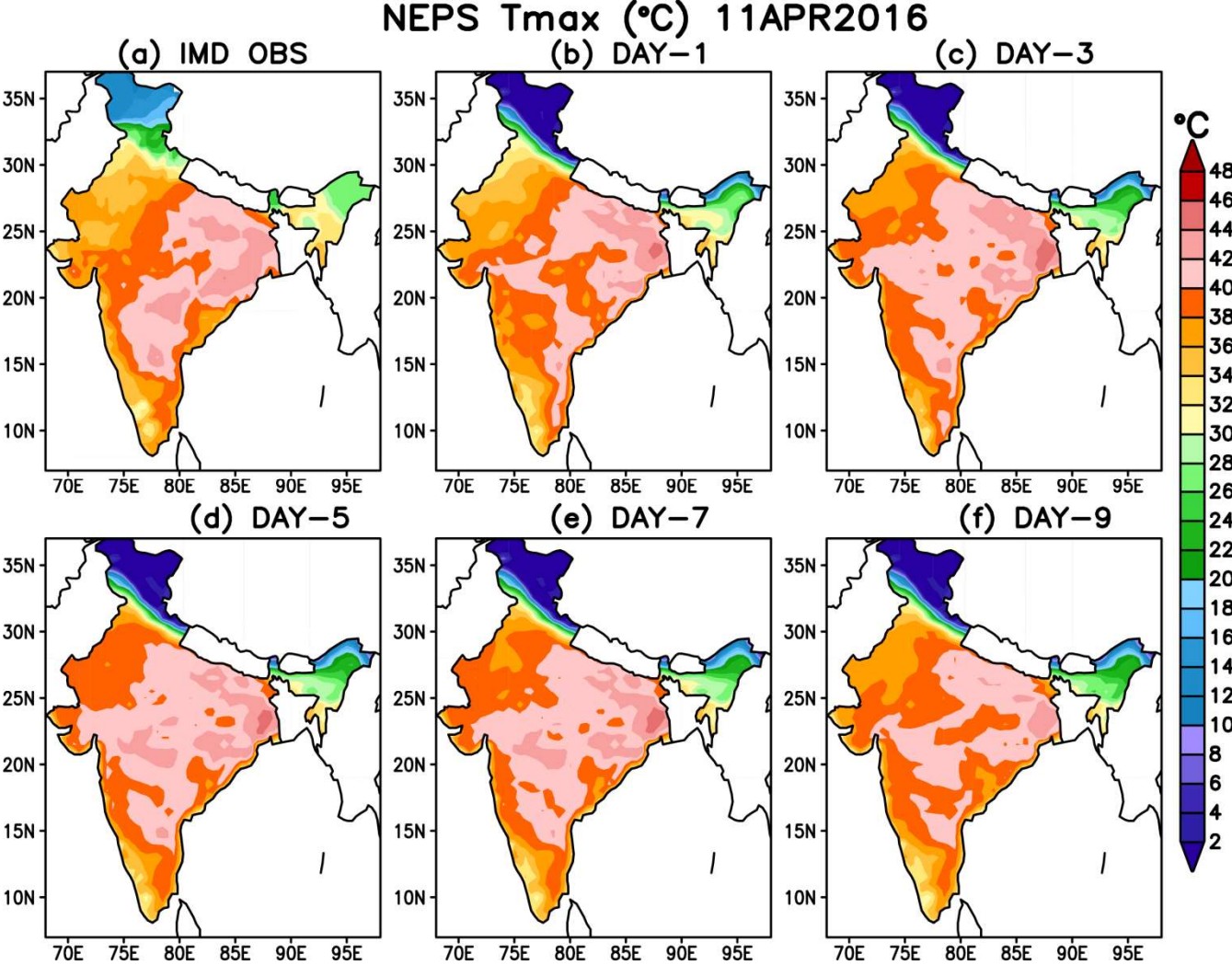

Figure 6: Spatial distributions of Observed *Tmax* (a) and NEPS forecast *Tmax* (b-f) prevailing heat waves over, MP, Odisha, AP, Telangana and some parts of Maharashtra on 11[th] April 2016. (NEPS stands for NCMRWF Ensemble Prediction System). [Observed data from India Meteorological Department and forecast data from NCMRWF]. Observed (Fig. 6a) *Tmax* ≥ 40°C is widespread over eastern India and large part of peninsula. NEPS forecasts (Fig. 6b-f) also show similar distribution.

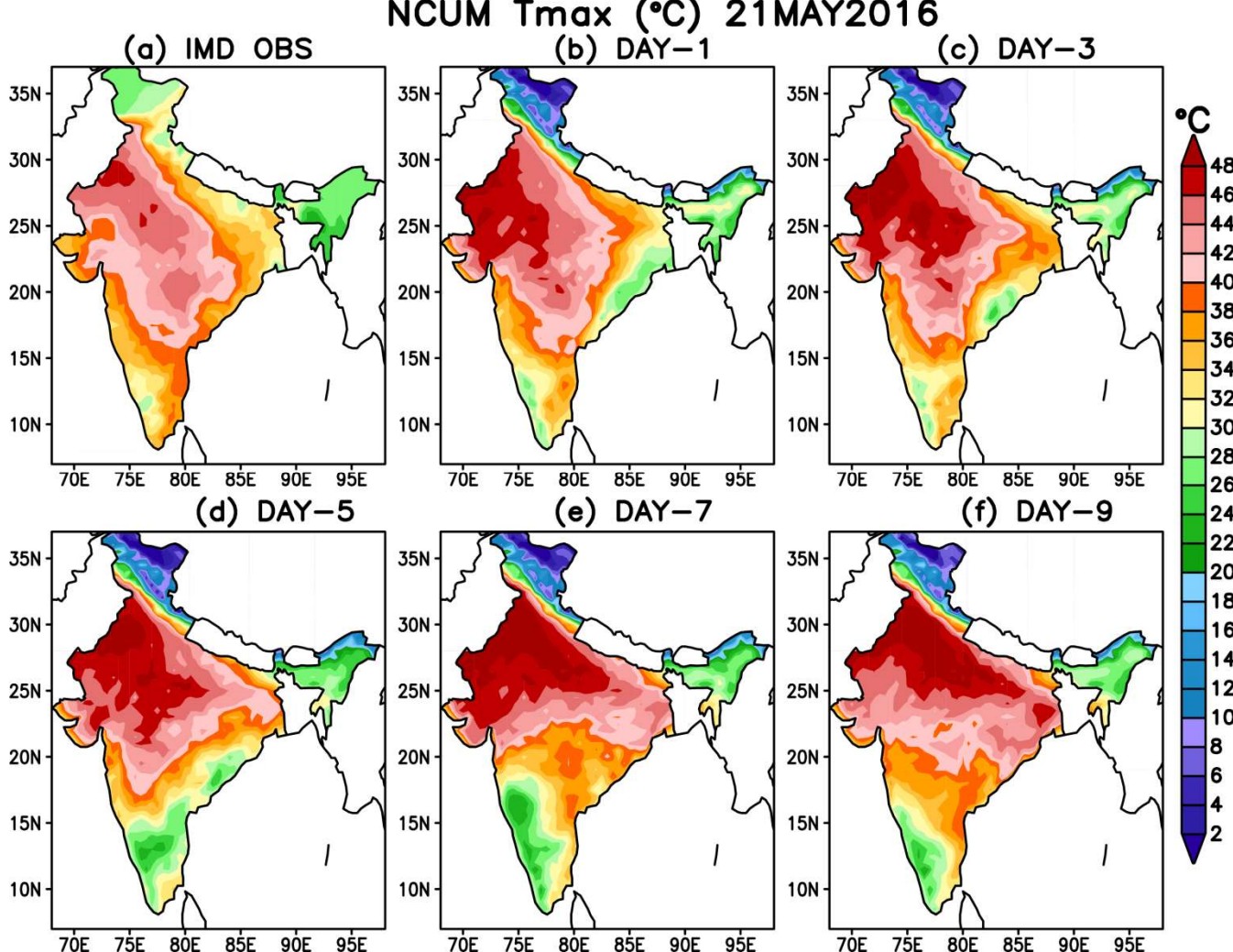

Figure 7: Spatial distributions of Observed *Tmax* (a) and NCUM forecast *Tmax* (b-f) prevailing heat waves over Rajasthan, MP, UP, Delhi, Haryana, Punjab and some parts of Maharashtra on 21[st] May 2016. (NCUM stand for NCMRWF Unified Model). [Observed data from India Meteorological Department and forecast data from NCMRWF]. Observed (Fig. 7a) *Tmax* ≥ 40°C is widespread over northwest and central India. NCUM forecasts (Fig. 7b-f) also show similar distribution. However, the forecasts indicate much stronger warming compared to observations. The *Tmax* ≥ 46°C is confined to a small region in the observations, while in the NCUM forecasts, it is seen widespread over northwest India.

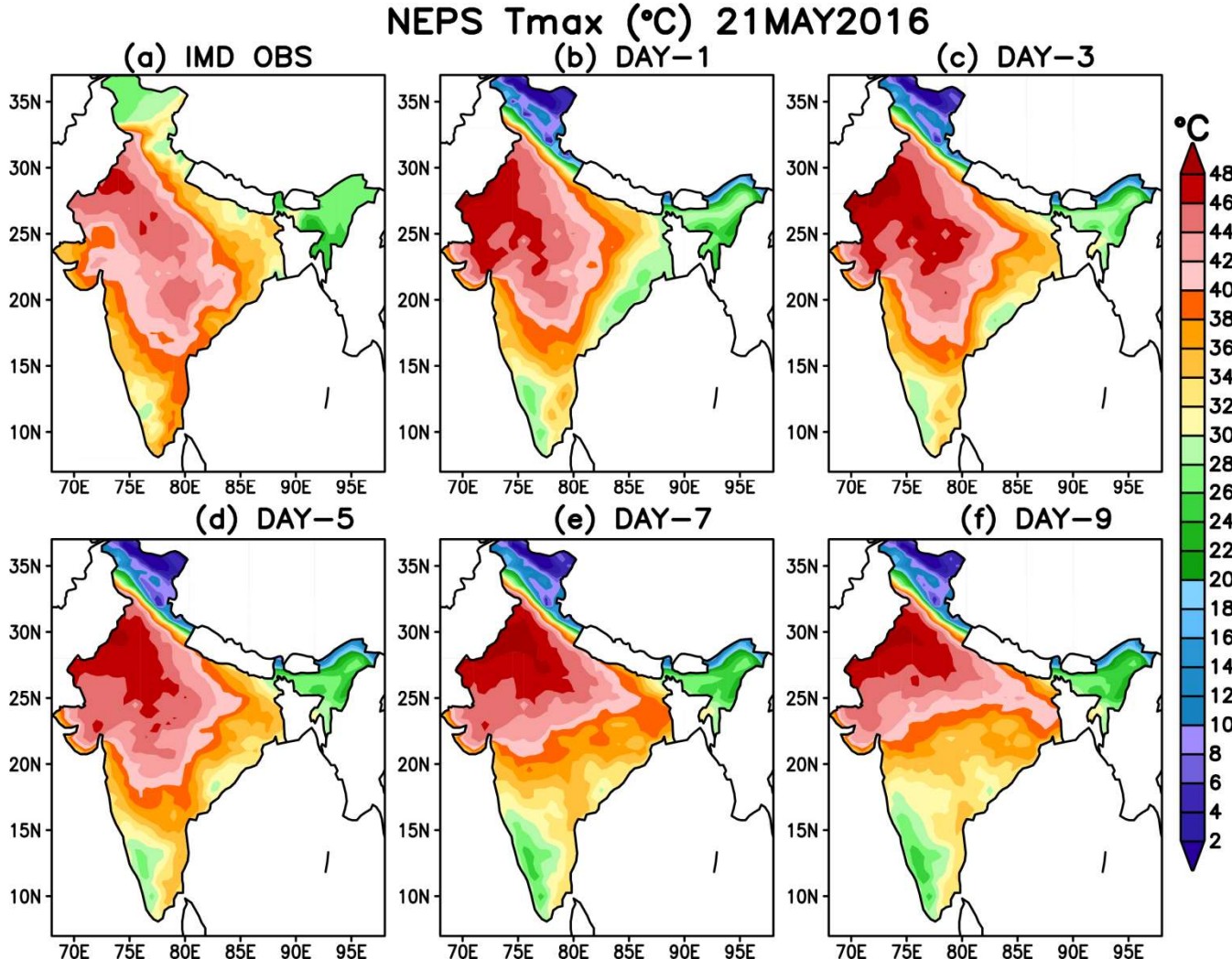

Figure 8: Spatial distributions of Observed *Tmax* (a) and NEPS forecast *Tmax* (b-f) prevailing heat waves over Rajasthan, MP, UP, Delhi, Haryana, Punjab and some parts of Maharashtra on 21[st] May 2016. (NEPS stands for NCMRWF Ensemble Prediction System). [Observed data from India Meteorological Department and forecast data from NCMRWF]. Observed (Fig. 8a) *Tmax* $\geq$ 40°C is widespread over northwest and central India. NEPS forecasts (Fig. 8b-f) also show similar distribution. However, the forecasts indicate much stronger warming compared to observations. The *Tmax* $\geq$ 46°C is confined to a small region in the observations, while in the NEPS forecasts; it is seen widespread over northwest India. Additionally, in NEPS day 7 and day 9 forecasts (Fig. 8e,f) *Tmax* $\geq$ 40°C is not extending to over peninsula and is seen stretching eastwards compared to observations (Fig. 8a).

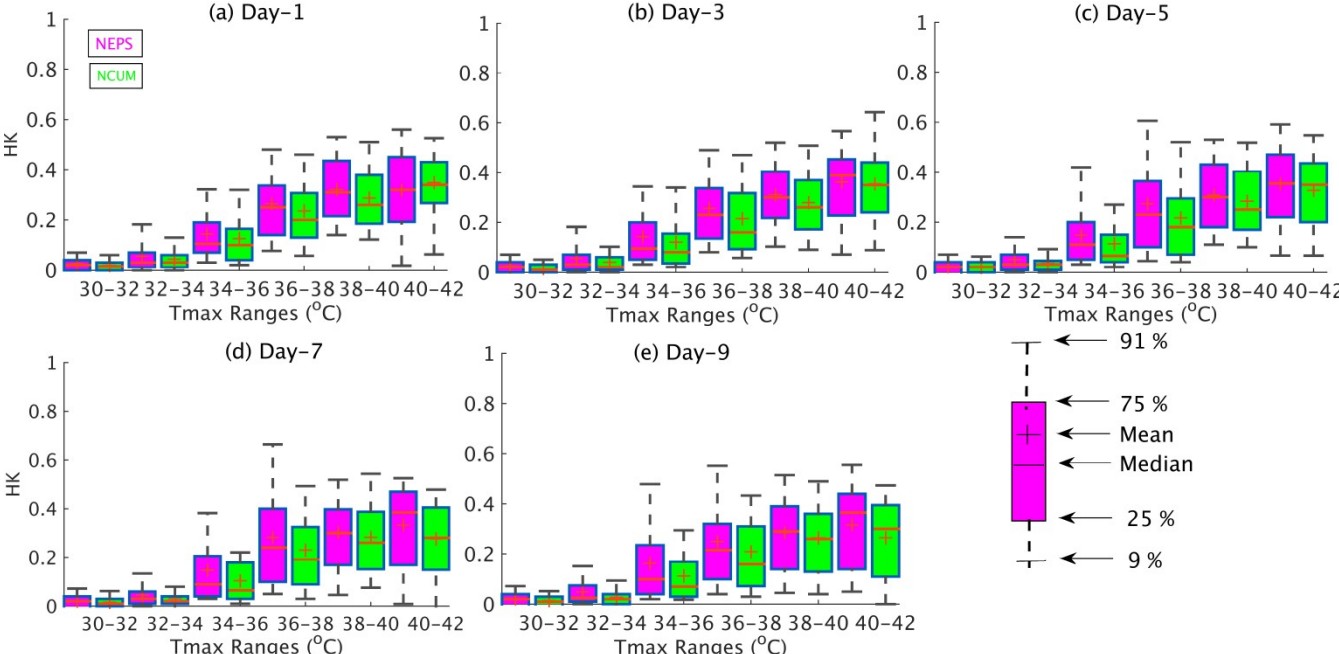

Figure 9: Box plots (a-e) for Hanssen **and** Kuipers (HK) scores for different temperature ranges (*Tmax*) NCUM and NEPS from March to May 2016. (NCUM stands for NCMRWF Unified Model and NEPS stands for NCMRWF Ensemble Prediction System). HK Score values are higher in NEPS forecasts particularly at high temperature ranges (>36 °C) and for all lead times compared to NCUM forecasts, suggesting improved performance of NEPS at higher temperature ranges.

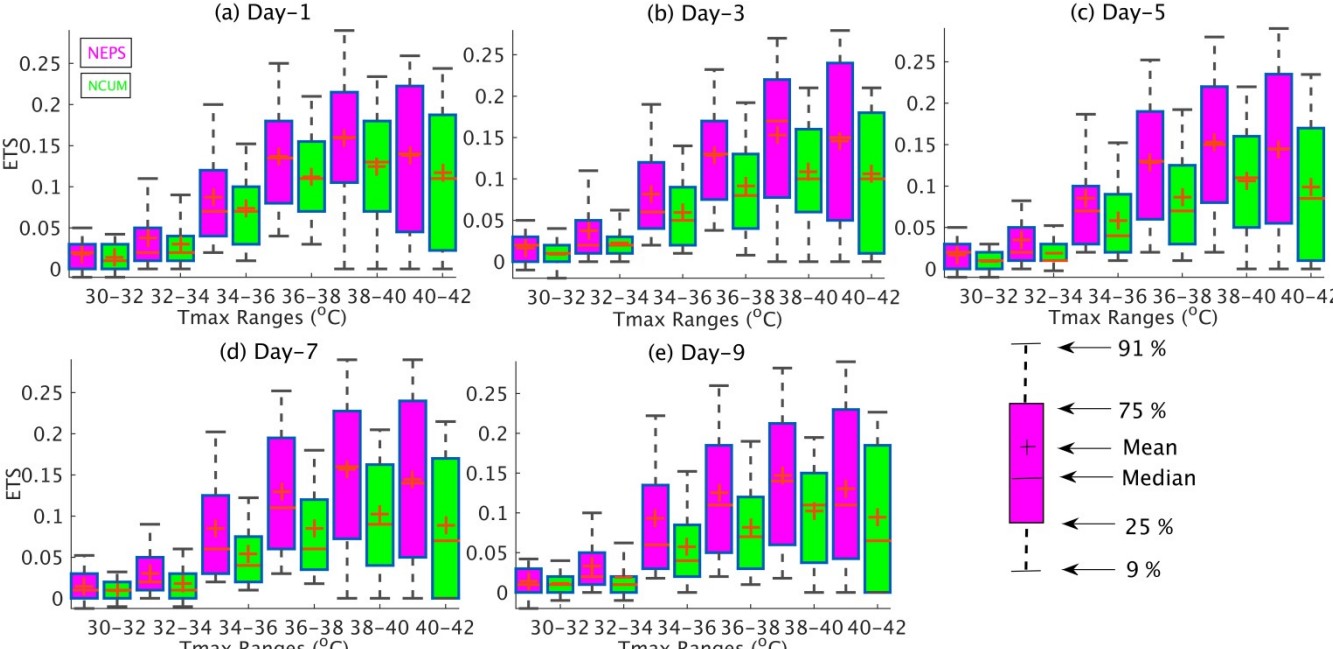

Figure 10: Box plots (a-e) for Equitable Threat Score (ETS) for NCUM and NEPS from March to May 2016. (NCUM stands for NCMRWF Unified Model and NEPS stands for NCMRWF Ensemble Prediction System). ETS values are higher in NEPS forecasts particularly at high temperature ranges (>36°C) and for all lead times compared to NCUM forecasts, suggesting improved performance of NEPS at higher temperature ranges.

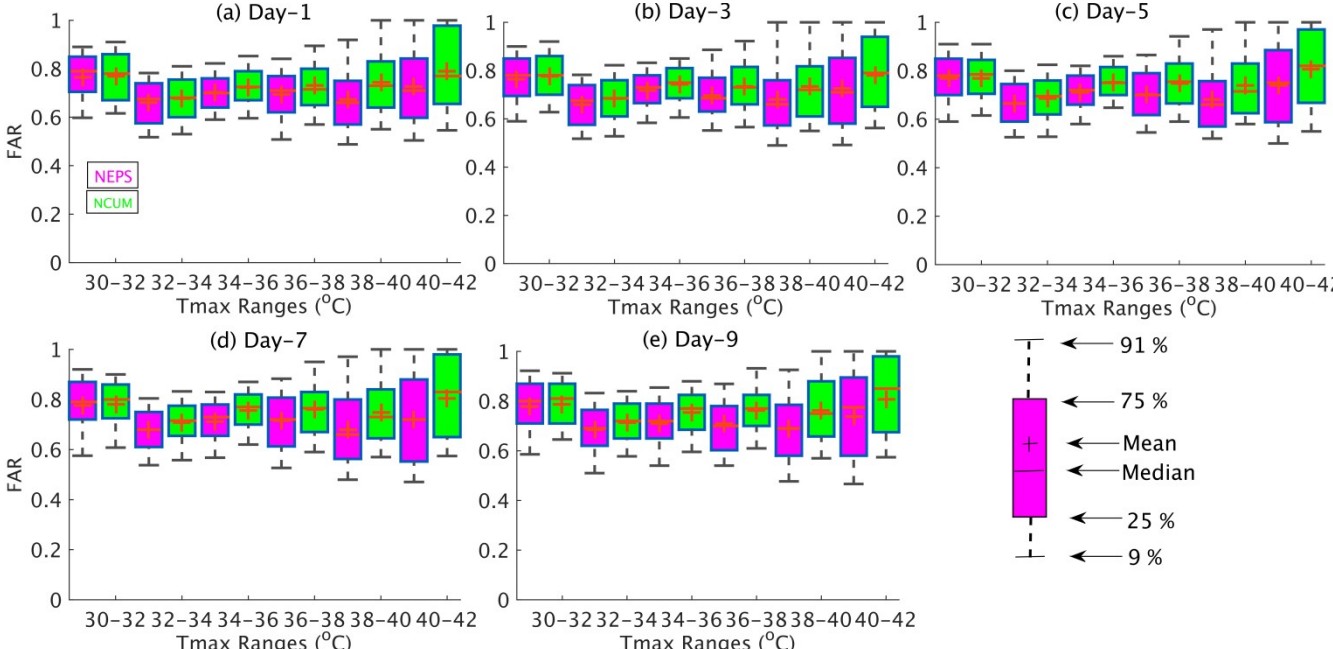

Figure 11: Box plots (a-e) for False Alarm Ratio (FAR) for NCUM and NEPS from March to May 2016. (NCUM stands for NCMRWF Unified Model and NEPS stands for NCMRWF Ensemble Prediction System). FAR values are lower in NEPS forecasts particularly at high temperature ranges (>36 °C) and for all lead times compared to NCUM forecasts, suggesting improved performance of NEPS at higher temperature ranges.

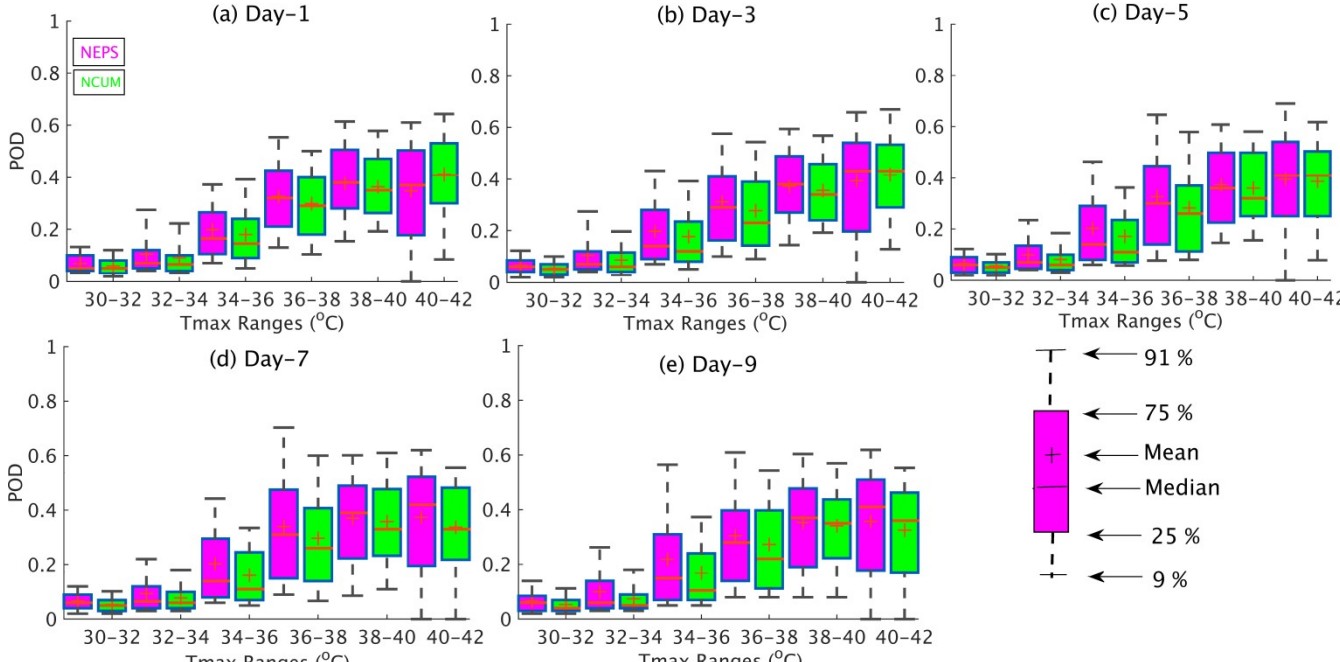

Figure 12: Box plots (a-e) for Probability of Detection (POD) for NCUM and NEPS from March to May 2016. (NCUM stands for NCMRWF Unified Model and NEPS stands for NCMRWF Ensemble Prediction System). POD values are higher in NEPS forecasts particularly at high temperature ranges (>36 °C) and for all lead times compared to NCUM forecasts, suggesting improved performance of NEPS at higher temperature ranges.

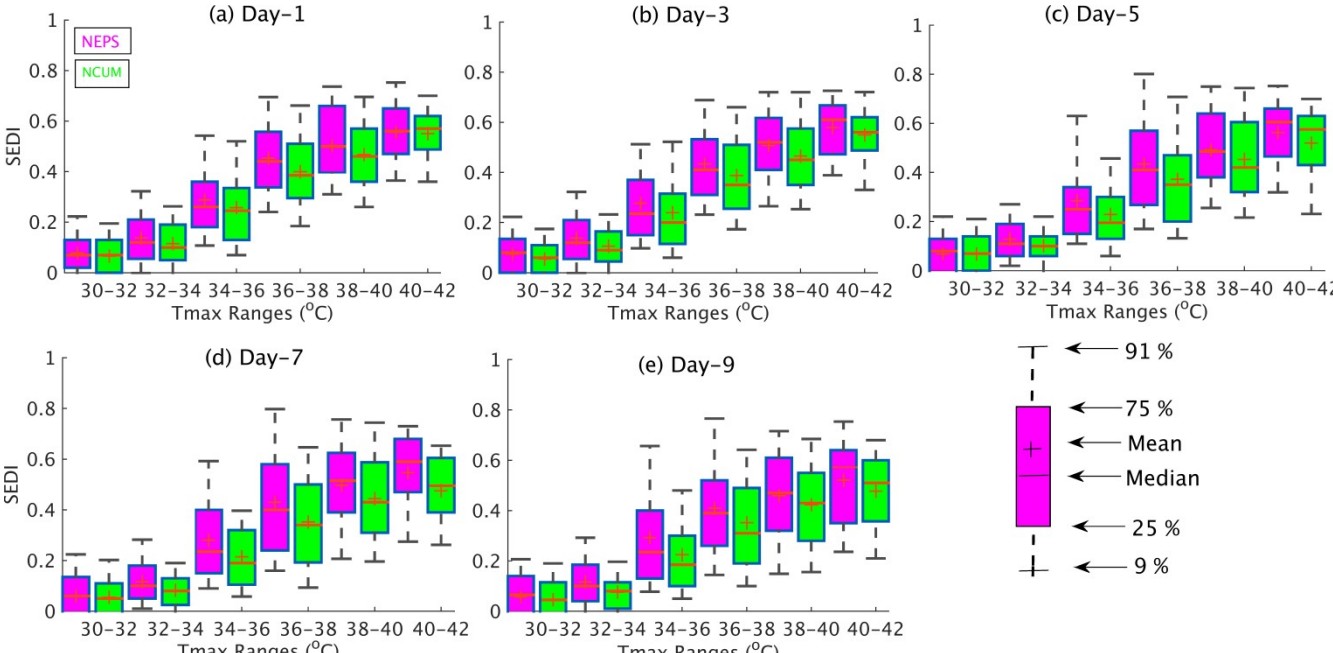

Figure 13: Box plots (a-e) for Symmetric Extremal Dependence Index (SEDI)) for NCUM and NEPS from March to May 2016. (NCUM stands for NCMRWF Unified Model and NEPS stands for NCMRWF Ensemble Prediction System). SEDI values are higher in NEPS forecasts for all temperature ranges and for all lead times compared to NCUM forecasts, suggesting improved performance of NEPS at all temperature ranges.

**Table 1: List of Abbreviations**

| AP | Andhra Pradesh |
|---|---|
| EPS | Ensemble Prediction Systems |
| ETKF | Ensemble Transform Kalman Filter |
| ETS | Equitable Threat Score |
| FAR | False alarm ratio |
| HK | Hanssen and Kuipers |
| IMD | Indian Meteorological Department |
| J&K | Jammu & Kashmir |
| MAM | March, April and May |
| MOGREPS | Met Office Global and Regional Ensemble Prediction System |
| MP | Madhya Pradesh |
| MSE | Mean Square Error |
| MSLP | Mean Sea Level Pressure |
| NCMRWF | National Centre for Medium Range Weather Forecasting |
| NCUM | NCMRWF Unified Model |
| NDC | National Data Centre |
| NEPS | NCMRWF Ensemble Prediction System |
| NWP | Numerical Weather Prediction |
| OPS | Observation Processing System |
| POD | Probability Of Detection |
| RMSE | Root Mean Square Error |
| SEDI | Symmetric Extremal Dependence Index |
| *Tmax* | Maximum Temperature |
| *Tmin* | Minimum Temperature |
| UK | United Kingdome |
| UM | Unified Model |
| UP | Uttar Pradesh |
| WMO | World Meteorological Organization |

**Table 2: Casualties reported during March to May 2016 due to prevailing heat waves over India**

| Month | State/ Region | No. of loss of lives | Total |
|-------|---------------|----------------------|-------|
| March | Maharashtra | 1 | 2 |
| | Kerala | 1 | |
| April | Odisha | **88** | 220 |
| | Telangana | 79 | |
| | AP | 40 | |
| | Maharashtra | 9 | |
| | Karnataka | 1 | |
| | Tamil Nadu | 1 | |
| May | Telangana | **200** | 273 |
| | Gujrat | 39 | |
| | Maharashtra | 34 | |

*(Data: Climate Diagnostic Bulletin of India, March 2016, April 2016 and May 2016, India Meteorological Department)*

Table 3: Month wise verification scores for *Tmax* > 40°C for NCUM and NEPS forecast for day 1 to day 9 lead times with India Meteorological Department (IMD) observed temperature.

| Month | Score | NCUM | | | | | NEPS | | | | |
|-------|-------|-------|-------|-------|-------|-------|-------|-------|-------|-------|-------|
| | | Day 1 | Day 3 | Day 5 | Day 7 | Day 9 | Day 1 | Day 3 | Day 5 | Day 7 | Day 9 |
| MAR | POD | 0.25 | 0.23 | 0.27 | 0.30 | 0.28 | 0.23 | 0.20 | 0.22 | 0.24 | 0.22 |
| | FAR | 0.81 | 0.71 | 0.75 | 0.75 | 0.79 | 0.49 | 0.54 | 0.53 | 0.53 | 0.43 |
| | ETS | 0.09 | 0.09 | 0.09 | 0.08 | 0.08 | 0.10 | 0.09 | 0.10 | 0.11 | 0.11 |
| | HK | 0.22 | 0.21 | 0.24 | 0.27 | 0.25 | 0.21 | 0.18 | 0.21 | 0.23 | 0.21 |
| | SEDI | 0.33 | 0.32 | 0.36 | 0.38 | 0.36 | 0.31 | 0.30 | 0.34 | 0.34 | 0.33 |
| APR | POD | 0.39 | 0.39 | 0.38 | 0.36 | 0.36 | 0.43 | 0.43 | 0.41 | 0.42 | - |
| | FAR | 0.66 | 0.65 | 0.66 | 0.66 | 0.66 | 0.62 | 0.61 | 0.62 | 0.61 | 0.62 |
| | ETS | 0.16 | 0.16 | 0.15 | 0.15 | 0.15 | 0.19 | 0.19 | 0.19 | 0.19 | 0.19 |
| | HK | 0.30 | 0.29 | 0.28 | 0.27 | 0.26 | 0.34 | 0.34 | 0.34 | 0.33 | 0.33 |
| | SEDI | 0.46 | 0.45 | 0.45 | 0.43 | 0.42 | 0.51 | 0.51 | 0.52 | 0.51 | 0.50 |
| MAY | POD | 0.30 | 0.30 | 0.28 | 0.26 | 0.24 | 0.32 | 0.34 | 0.31 | 0.31 | 0.27 |
| | FAR | 0.70 | 0.71 | 0.72 | 0.74 | 0.75 | 0.67 | 0.69 | 0.70 | 0.71 | 0.75 |
| | ETS | 0.12 | 0.11 | 0.11 | 0.10 | 0.09 | 0.14 | 0.14 | 0.13 | 0.12 | 0.10 |
| | HK | 0.22 | 0.22 | 0.21 | 0.19 | 0.17 | 0.25 | 0.26 | 0.24 | 0.23 | 0.19 |
| | SEDI | 0.39 | 0.38 | 0.36 | 0.33 | 0.30 | 0.43 | 0.43 | 0.40 | 0.39 | 0.33 |

(NCUM stand for NCMRWF Unified Model and NEPS stands for NCMRWF Ensemble Prediction System)

