# Peer review of "Verification of Pre-Monsoon Temperature Forecasts over India during 2016 with focus on Heatwave Prediction"

_Natural Hazards and Earth System Sciences, 2016_

## Referee Comment (RC1) · Anonymous Referee #1 · 25 Nov 2016

**Review of manuscript "Verification of Pre-Monsoon Temperature Forecasts over India during 2016 with focus on Heat Wave Prediction"     By Harvir Singh, Kopal Arora, Raghvendra Ashrit and EN Rajagopal (Ms No: nhess-2016-264**

**General Comments:**

As the title suggests this paper investigates the extreme heat wave events over India using a high-resolution numerical weather prediction and its ensemble forecast in union with the classical statistical scores to serve the verification purposes. The authors claim that the results indicate an appreciable competence of the ensemble forecasting detecting extreme heat wave events as compared to deterministic forecast. On the whole the scientific significance and the presentation quality are good.

**Specific Comments:**

(i) The verifications are based on extreme heat events for only one year, the authors could consider a few more years to support their results

(ii) The authors have used gridded data. This would have definitely suppressed the extreme station temperature values

(iii) On Page 2, Kothawale (2005) and IPCC (2013) have been cited but not listed under references

(iv) Page 6 : The y-axis in each of the figs (is this applicable to all figures ?)

(v) Page 7: Line 23, mention is made of Table-1, but this table lists the abbreviations used

(vi) Page 7 last 3 lines: Authors mention spatial distribution ----- but Fig. 8 and 9 show box plots

(vii)     Page 8: first Fig 11 is referred, then 10, then 9 and the 8 ??? Please follow sequence

(viii)     Page 8 line 7 Mention of ETS plots (Fig.10) is made but this fig contains plots for FAR

(ix) Similarly Fig. 9 are ETS plots but in text something else is mentioned (page 8)

(x) Page 8 line 23: Mention is made of SEDI score plot – fig number not mentioned

(xi) Several repetitions

---

## Referee Comment (RC2) · Anonymous Referee #2 · 20 Dec 2016

General Comments ——————— This research on the predictability of heatwaves over India is of huge potential interest and humanitarian benefit given the loss of life suffered in these events. The authors demonstrate predictability in both deterministic and ensemble forecasts, with ensembles showing some marginal increase in skill. The manuscript is reasonably well organised but there are a lot of inconsistencies and technical errors that need addressed before this manuscript can be considered for final publication (see below).

Specific Comments ———————-

1. How much of the skill in predicting the heatwaves comes from persisting a heatwave already present in the initial conditions? How does the model perform when the

heatwave evolves within the forecast range (e.g. Beyond days 2-3).

2. Synoptic evolution in heatwave case studies - It would have been good to also see the prevailing synoptic conditions and larger-scale flow conditions associated with these heatwaves ( e.g. MSLP or low level winds) in both observations/analysis and deterministic and EPS (ensemble mean) forecasts. Perhaps also the time series of temperatures (deterministic and EPS members (at day 2, 5, 7), and Observations) over a specific region (e.g. Rajasthan) during one of the heatwave events would also give the reader a more physical feel for the predictability that is difficult to get just from the verification metrics alone. This is achieved to some extent by snapshots in Figs 4-7.

3. Could the authors provide more detail on how the various categorical scores are calculated for the EPS. Are the scores based on the ensemble mean vs. observations or do they use all 44 individual ensemble members to construct a score?

4. Page 6, lines 11-12 - "Deterministic forecast hardly shows any variation in either of the considered days and illustrates quasi-stationary characteristics of the deterministic forecast from Day-1 through Day-10 forecast". I don't really understand this or know which figure/result it is referencing. Can the authors clarify.

5. Figure 1 suggests that the deterministic forecasts (and to a lesser extent the EPS) underpredicts the frequency of heatwaves compared to observations over Indian land points. This appears to be inconsistent with later discussions around figures 2 and 3 which suggest that the deterministic and EPS over predict the number of heatwave days (>40) compared to the Observations? Can the authors explain this inconsistency?

6. In Fig 6. the NCUM and to lesser extent the NEPS forecasts show a growing warm bias over NW India with FC range. Do the authors have any physical explanation for this bias (e.g. soil moisture initialisation, model systematic errors in circulation?)

7. Predictability of heatwaves - In the summary the authors state "Unless the atmosphere is in a highly predictable state, we should not expect an ensemble to forecast extreme events with a high probability". It would be good to see some discussion of whether these heatwave events are highly predictable (e.g. links to large scale flow anomalies), given they seemed to be predictable several days ahead? Was the ensemble spread of Tmax smaller or larger than normal in these heatwaves?

8. Are there plans to use these EPS predictions of heatwaves to give warnings to the public? Perhaps some discussion in summary?

Technical corrections ——————————

This manuscript suffers from a lot of technical errors and inconsistencies that make it difficult to read. Some of these relate to English useage but many are just errors that are easily corrected. I have listed the main errors below

1. A number of variations on the word "heatwave" appear in the manuscript (Heat wave, Heat Wave, heat wave and heatwave). Suggest authors provide a consistent spelling (e.g. heatwave).

2. Authors also refer to "deterministic models" and "ensemble models". This should be replaced with "deterministic forecasts" and "ensemble forecasts" throughout the text as both actually use the same model (UM).

3. Page 1, Line 9 removed "the" in this sentence - here we investigate extreme events (heatwaves)

4. Page 1, Line 22 - replace "...prediction the extreme events" with "...prediction of extreme events"

5. Page 1, Line 22 - I don't understand the sentence "Extreme Weather events comprehend non-linear interactions..."

6. Page 1, Line 30 - simplify this sentence "Based on multiple perturbed initial conditions, ensemble approach samples the errors in the initial..." to "It is based on

7. Page 2, line 1 - remove the first reference to Sarkar et. al., 2009, as it is repetitive.

8. Page 2 line 2 - Replace "Met office" with "Met Office"

9. Page 2 line 14 - replace "0.85 0°C" with "0.85 °C"

10. Page 2 line 15 - don't understand how Molteni et. al. (1996) could be used as reference for a warming trend covering 1880-2012!

11. Page 2 line 17 - assume that the annual mean temperature of 0.42 C per 100 years refers to the globally averaged temperatures. This should be made clear.

12. Page 2 line 21 - this paragraph begins with a sentence "Another study..." but the reference at the end of the sentence is Arora et. al. (2009) which was the same study discussed in the previous paragraph.

13. Page 2 Line 24 - not sure what "recently reiterated" means?

14. Page 2 line 28 - "sales" should read "scales"

15. Page 2 line 29 - the sentence "...using ensemble forecast forecasts both, deterministic and ensemble forecasting." is very convoluted, can I suggest "...using both ensemble and deterministic forecasts"

16. Page 3 line 9 - delete "and" in the following "...adopt and the most compatible score type"

17. Page 3 line 11 - this sentence is very repetitive.

18. Page 3 line 23 - remove "... which was 1°x1° resolution a few years earlier over Indian land area." As it is irrelevant for this study.

19. Page 3 Line 32 - replace "operational NCMRWF" with "operational at NCMRWF

20. Page 4 line 8 - replace "...MET Office" with "...Met Office MOGREPS system (Bowler et. al. 2008)" where reference is Bowler, N. E., Arribas, A., Mylne, K. R., Robertson, K. B. and Beare, S. E. (2008), The MOGREPS short-range ensemble prediction system. Q.J.R. Meteorol. Soc., 134: 703–722. doi:10.1002/qj.234

21. Page 4 Line 14 - replace "Uncertainty in forecasting model..." with "Uncertainty in the forecasting model..."

22. Page 4 line 16 - Remove this line as it is repetitive (see line 4-5 on this page which says the same thing)

23. Page 5 line 8 - Heidke skill score mentioned but not defined or used. Remove this reference?

24. Page 5 line 26 - replace "...efficiency" with "...capability"?

25. Page 6 line 9 - replace "... the figures (Fig. 5) and (Fig. 4)." with "Fig. 5 and Fig. 4."

26. Page 6 line 11 - use "The deterministic forecast..."

27. Page 6 lines 11-12 Replace "...any variation in either of the considered days and illustrates quasi-stationary characteristics of the deterministic forecast from Day-1 through Day-10 forecast" with "... any variation in either of the days and illustrates quasi-stationary characteristics from Day-1 through Day-10"

28. Page 6 line 13 - Remove "...and vary in not so distinctive fashion".

29. Page 6 line 15 - "Fig. (2.)" should read "Fig.2"

30. Page 6 line 15 - Remove "..(Tmax).."

31. Page 8 line 7 - Fig 10. should read Fig 9.

32. Page 8 line 18 - Fig. 9 should read Fig. 10.

33. Page 8 line 21 - missing end parentheses ")"

34. Page 8 line 23 - missing figure number.

Figures and tables ———————— 1. Figure 2 and 3 - the colour bar is labelled °C

When the quantity is a count. 2. Table 2 title - "Causalities" should read "Casualties

Please also note the supplement to this comment:
http://www.nat-hazards-earth-syst-sci-discuss.net/nhess-2016-264/nhess-2016-264-RC2-supplement.pdf

―――――――――――――――

---

## Author Comment (AC1) · 27 Dec 2016

We are thankful to the editor and the anonymous reviewer(s) for their helpful suggestions which have helped us to improve the quality of the paper to a great extent. We have tried to incorporate as many of their suggestions as possible.

Replies to the Reviewer #1 comments

1. The verifications are based on extreme heat events for only one year, the authors could consider a few more years to support their results.

Reply:The suggestion by the reviewer is very valid. For the present study the data from the two modes is available only from 2016. Ensemble based forecasts in realtime using

the NEPS started in November 2015 at NCMRWF. For a robust and conclusive results it is necessary that the study be based higher number of cases. This will be carried out in future.

2. The authors have used gridded data. This would have definitely suppressed the extreme station temperature values.

Reply:The temperature data from the stations distribution are discussed in the paper which are used to obtain the gridded Tmax and Tmin data. It is indeed likely that some of the station extremes are smoothed out in the gridded data. It should also be noted that the stations data network is sparse 395 and often there are missing values. Gridded data field provides a continuous and gap free data to workwith.

Replies to the Reviewer #2 comments

1. How much of the skill in predicting the heatwaves comes from persisting a heat-wave already present in the initial conditions? How does the model perform when the heatwave evolves within the forecast range (e.g. Beyond days 2-3).

Reply: Extreme events like heatwaves are rare in nature and here we provided a gen-eral view of the two particular heatwave events (11 April & 21 May). From our ex-perience as well as the forecast for the post heatwave event days, we can state that the skill of predicting an event with the initial conditions of no indication of severity is comparatively lower than when the signature is present in the initial conditions. Even before the event, there is some signature of it as can be seen in the figure. The overall prediction of warm conditions is nicely predicted but at closer lead times, the events are better predicted. Same can be seen in the box and whisker plots for ETS (and rest of the score plots as well). For instance, the skill of NEPS does not fall drastically from Day-2 to Day-7 and thus depicts a reasonable skill. So, overall the NEPS specifically, has a good skill in predicting the extreme event and is relatively robust.

2. Synoptic evolution in heatwave case studies - It would have been good to also

see the prevailing synoptic conditions and larger-scale flow conditions associated with these heatwaves ( e.g. MSLP or low level winds) in both observations/analysis and deterministic and EPS (ensemble mean) forecasts. Perhaps also the time series of temperatures (deterministic and EPS members (at day 2, 5, 7), and Observations) over a specific region (e.g. Rajasthan) during one of the heatwave events would also give the reader a more physical feel for the predictability that is difficult to get just from the verification metrics alone. This is achieved to some extent by snapshots in Figs 4-7.

Reply: Thank you for your insightful comment. As per your suggestion, we are adding a figure illustrating synoptic systems (both, MSLP & low-level winds) for the heatwave event considered in the present work (Dated:20160521). We can see that the monsoon heat low shown by low MSLP values over NW Indian and adjoining Pakistan is an important semi-permanent system during the pre-monsoon season. The low MSLP values and high temperatures associated with that create strong land-sea temperature and pressure gradient in the lower troposphere which is crucial for onset and advance of monsoon. As can be seen in the figure below, during this pre-monsoon month, the low pressure is accompanied by the westerly and north-westerly winds and heatwaves over the Indian and the neighboring countries. In the figure, we see it mainly occurring over the central India.

3. Could the authors provide more detail on how the various categorical scores are calculated for the EPS. Are the scores based on the ensemble mean vs. observations or do they use all 44 individual ensemble members to construct a score?

Reply: Computation of the scores is based on the ensemble mean (44 members). An ensemble mean is first computed from each member which is then treated as another model and is further used to obtain the scores. It is known that the ensemble mean has a higher skill than the deterministic forecast especially in the upper air fields (500 hPa) (cite: Ton Hamil et. al) and similar observation is justifiable for the low-level fields as well (fig: score plots).

4. Page 6, lines 11-12 - "Deterministic forecast hardly shows any variation in either of the considered days and illustrates quasi-stationary characteristics of the deterministic forecast from Day-1 through Day-10 forecast". I don't really understand this or know which figure/result it is referencing. Can the authors clarify.

Reply: Suitably modified and referred to a figure. "The deterministic forecast shows lesser variation as compared to the ensemble forecast on either of the considered days and possibly show a quasi-stationary characteristic of the deterministic forecast from Day-1 through Day-10 forecast (Fig. 2, Fig. 3)"

5. Figure 1 suggests that the deterministic forecasts (and to a lesser extent the EPS) underpredicts the frequency of heatwaves compared to observations over Indian land points. This appears to be inconsistent with later discussions around figures 2 and 3 which suggest that the deterministic and EPS over predict the number of heatwave days (>40) compared to the Observations? Can the authors explain this inconsistency?

Reply: The figure was prepared to choose ranges of the verification metrics and does not serve a purpose to indicate any sort of over or under prediction. This is because the figure represents the "fraction" of the total number of days and the grid points (i.e. counts/92X2686 (days X grids)). The denominator includes all the grid points with or without the Tmax > 40C.

6. In Fig 6. the NCUM and to lesser extent the NEPS forecasts show a growing warm bias over NW India with FC range. Do the authors have any physical explanation for this bias (e.g. soil moisture initialization, model systematic errors in circulation?)

Reply: In 21 May case, warming is increasing drastically for both, NEPS and NCUM. This is not based on one initial condition and include several different initial conditions. We have error growth and warm bias In the present study the impact of soil moisture feedback is not attempted. The land surface scheme involves soil moisture data assimilation using extended Kalman Filter technique. The soil moisture analysis prepared based on screen level humidity and temperature observations and ASCAT

surface soil wetness observations from MetOP-A satellite (C- band, Level2 product). Systematic errors in circulation have been widely and extensively studied and documented for monsoon (JJAS) season. Typically for the pre-monsoon conditions such detailed analysis would be useful and will be taken up as a follow up of this study.

7. Predictability of heatwaves - In the summary the authors state "Unless the atmosphere is in a highly predictable state, we should not expect an ensemble to forecast extreme events with a high probability". It would be good to see some discussion of whether these heatwave events are highly predictable (e.g. links to large scale flow anomalies), given they seemed to be predictable several days ahead? Was the ensemble spread of Tmax smaller or larger than normal in these heatwaves?

Reply: As stated earlier in response to the second point, the extreme events are rare which offeres a small sample size, thereby making their predictability and verification difficult as such. However, signature of the events are noticeable in the synoptic systems, a few days ahead of the event (ex. Wind patterns and MSLP, fig).

8. Are there plans to use these EPS predictions of heatwaves to give warnings to the public? Perhaps some discussion in summary?

Reply: This preliminary study indicates potential skill in forecasting heatwave conditions. With the use of suitable calibration/downscaling and bias correction methods, these forecasts of heatwaves could be useful for the forecasters at operational agency Indian Meteorological Department.

Apart from these specific comments we have incorporated all the technical errors/comments in the manuscripts.
* * *
[Figure]

Fig. 1.

---

## Author Response (AR1)

Title: **Verification of Pre-Monsoon Temperature Forecasts over India during 2016 with focus on Heat Wave**

**Prediction**

**NHESS-2016**

**We are thankful to the editor and the reviewer for their helpful suggestions which have helped us to improve the quality of the paper to a great extent. We have tried to incorporate as many of their suggestions as possible.**

**Reviewer #1 Specific Comments**

1. *The verifications are based on extreme heat events for only one year, the authors could consider a few more years to support their results*

   **Reply 1**: The suggestion by the reviewer is very valid. For the present study the data from the two models is available only from 2016. Ensemble based forecasts in real time using the NEPS started in November 2015 at NCMRWF. For robust and conclusive results it is necessary that the study be based higher number of cases. This will be carried out in future.

2. *The authors have used gridded data. This would have definitely suppressed the extreme station temperature values*

   **Reply 2**: The temperature data from the stations distribution are discussed in the paper which is used to obtain the gridded *Tmax* and *Tmin* data. It is indeed likely that some of the station extremes are smoothed out in the gridded data. It should also be noted that the stations data network is sparse 395 and often there are missing values. Gridded data field provides a continuous and gap free data to work with.

**Reviewer #1 Technical corrections**

3. *On Page 2, Kothawale (2005) and IPCC (2013) have been cited but not listed under references*

   **Reply 3**: Thanks for pointing out this error. The first reference Kothawale et al (2005) is removed and is replaced by more recent study by same author Kothawale et al (2010) on *page 2 line 20*. The IPCC(2013) is included in the references on page 12 line 4.

   Additionally, the other reference of Arora et al (2009) is eliminated in the text on page 2(*line 23*) and in the list of references.

4. *Page 6 : The y-axis in each of the figs (is this applicable to all figures ?)*

**Reply 4**: No. This refers to two panels in Figure 1. The text on page 6 near line 26 has been modified as-

The panels in Fig. 1a,b  show the observed and forecast (Day-3) frequency distribution for *Tmax* and *Tmin*.

5. *Page 7: Line 23, mention is made of Table-1, but this table lists the abbreviations used*

   **Reply 5**: The text is corrected on page 8 line 9.

6. *Page 7 last 3 lines: Authors mention spatial distribution ----- but Fig. 8 and 9 show box plots*

   **Reply 6**: The discussion on two heatwave cases presented in sections 4.3.1 (and 4.3.2)  are based on Figs 5,6 (and 7,8) respectively. The figure numbers are correctly represented in the revised manuscript on page 8 in line 31 and on page 9 line 12.

7. *Page 8: first Fig 11 is referred, then 10, then 9 and the 8 ??? Please follow sequence*

   **Reply 7**: The text on page 9 (line 12 and 3) is corrected to refer to Figs 7&8.

8. *Page 8 line 7 Mention of ETS plots (Fig.10) is made but this fig contains plots for FAR*

   **Reply 8**: In the revised manuscript on page 9 line 19, the ETS is discussed using Fig 10.

9. *Similarly Fig. 9 are ETS plots but in text something else is mentioned (page 8)*

   **Reply 9**: The change is reflected on the page 9 line 18, now it is HK scores.

10. *Page 8 line 23: Mention is made of SEDI score plot – fig number not mentioned*

    **Reply 10**: In the revised manuscript the Fig 13 showing SEDI is correctly referred on page 10 line 5.

11. *Several repetitions*
    **Reply 12**: Thanks for bringing this to our notice. We have taken extra care to avoid the repetitions in the manuscript.

1. *How much of the skill in predicting the heatwaves comes from persisting a heatwave already present in the initial conditions? How does the model perform when the heatwave evolves within the forecast range (e.g. Beyond days 2-3).*

**Reply 1**: Extreme events like heat waves are rare in nature and here we provided a general view of the two particular heat wave events (11 April & 21 May). From our experience as well as the forecast for the post heat wave event days, we can state that the skill of predicting an event with the initial conditions of *no indication* of severity is comparatively *lower* than when the signature is present in the initial conditions.
Even before the event, there is some signature of it as can be seen in the figure (5, 6, 7 & 8). The overall prediction of warm conditions is nicely predicted but at closer lead times, the events are better predicted. Same can be seen in the box and whisker plots for ETS (and rest of the score plots as
well). For instance, the skill of NEPS does not fall drastically from Day-2 to Day-7 and thus depicts a reasonable skill. So, overall the NEPS specifically, has a good skill in predicting the extreme event and is relatively robust.

2. *Synoptic evolution in heatwave case studies - It would have been good to also see the prevailing synoptic conditions and larger-scale flow conditions associated with these heatwaves ( e.g. MSLP or low level winds) in both observations/analysis and deterministic and EPS (ensemble mean) forecasts. Perhaps also the time series of temperatures (deterministic and EPS members (at day 2, 5, 7), and Observations) over a specific region (e.g. Rajasthan) during one of the heatwave events would also give the reader a more physical feel for the predictability that is difficult to get just from the verification metrics alone. This is achieved to some extent by snapshots in Figs 4-7.*

**Reply 2:** Thank you for your insightful comment. As per your suggestion, we are adding a figure illustrating synoptic systems (both, MSLP & low-level winds) for the heat wave event considered in the present work (Dated: 20160521). We can see that the monsoon heat low shown by low MSLP values over NW Indian and adjoining Pakistan is an important semi-permanent system during the pre-monsoon season. The low MSLP values and high temperatures associated with that create strong land-sea temperature and pressure gradient in the lower troposphere which is crucial for onset and advance of monsoon. As can be seen in the figure below, during this pre-monsoon month, the low pressure is accompanied by the westerly and north-westerly winds and  heat waves over the Indian and the neighboring countries. In the figure, we see it mainly occurring over the central India.

[Figure]

3. *Could the authors provide more detail on how the various categorical scores are calculated    for the EPS. Are the scores based on the ensemble mean vs. observations or do they use all 44 individual ensemble members to construct a score?*

**Reply 3**: Computation of the scores is based on the ensemble mean (44 members). An ensemble mean is first computed from each member which is then treated as another model and is further used to obtain the scores. It is known that the ensemble mean has a higher skill than the deterministic forecast especially in the upper air fields (500 hPa) (cite: Ton Hamil et. al) and similar observation is justifiable for the low-level fields as well (fig: score plots).

4. *Page 6, lines 11-12 - "Deterministic forecast hardly shows any variation in either of the considered days and illustrates quasi-stationary characteristics of the deterministic forecast from Day-1 through Day-10 forecast". I don't really understand this or know which figure/result it is referencing. Can the authors clarify.*

**Reply 4:** Suitably modified and change is reflected on page 7 line 3.

5. *Figure 1 suggests that the deterministic forecasts (and to a lesser extent the EPS) underpredicts the frequency of heatwaves compared to observations over Indian land points. This appears to be inconsistent with later discussions around figures 2 and 3 which suggest that the deterministic and EPS over predict the number of heatwave days (>40) compared to the Observations? Can the authors explain this inconsistency?*

**Reply 5:** The figure was prepared to choose ranges of the verification metrics and does not serve a purpose to indicate any sort of over or under prediction. This is because the figure represents the "fraction" of the total number of days and the grid points (i.e. counts/92X2686 (days X grids)). The denominator includes all the grid points with or without the Tmax > 40C.

6. *In Fig 6. the NCUM and to lesser extent the NEPS forecasts show a growing warm bias over NW India with FC range. Do the authors have any physical explanation for this bias (e.g. soil moisture initialisation, model systematic errors in circulation?)*

**Reply 6:** In 21 May case, warming is increasing drastically for both, NEPS and NCUM. This is not based on one initial condition and includes several different initial conditions. We have error growth and warm bias in the present study the impact of soil moisture feedback is not attempted. The land surface scheme involves soil moisture data assimilation using extended Kalman Filter technique. The soil moisture analysis prepared based on screen level humidity and temperature observations and ASCAT surface soil wetness observations from MetOP-A satellite (C- band, Level2 product). Systematic errors in circulation have been widely and extensively studied and documented for monsoon (JJAS) season. Typically for the pre-monsoon conditions such detailed analysis would be useful and will be taken up as a follow up of this study.

7. *Predictability of heatwaves - In the summary the authors state "Unless the atmosphere is in a highly predictable state, we should not expect an ensemble to forecast extreme events with a high probability". It would be good to see some discussion of whether these heatwave events are highly predictable (e.g. links to large scale flow anomalies), given they seemed to be predictable several days ahead? Was the ensemble spread of Tmax smaller or larger than normal in these heatwaves?*

**Reply 7:** As stated earlier in response to the second point, the extreme events are rare which offers' a small sample size, thereby making their predictability and verification difficult as such. However, signature of the events are noticeable in the synoptic systems, a few days ahead of the event (ex. Wind patterns and MSLP, fig).

8. *Are there plans to use these EPS predictions of heatwaves to give warnings to the public? Perhaps some discussion in summary?*

**Reply 8:** This preliminary study indicates potential skill in forecasting heat wave conditions. With the use of suitable calibration/downscaling and bias correction methods, these forecasts of heat waves could be useful for the forecasters at operational agency Indian Meteorological Department.

**Technical corrections**

This manuscript suffers from a lot of technical errors and inconsistencies that make it difficult to read. Some of these relate to English usage but many are just errors that are easily corrected. I have listed the main errors below

1. *A number of variations on the word "heatwave" appear in the manuscript (Heat wave, Heat Wave, heat wave and heatwave). Suggest authors provide a consistent spelling (e.g. heatwave).*
   **Reply 1:** Thanks for bringing this to our notice; we have incorporated a consistent word *"heatwave"* in the entire revised manuscripts.
2. *Authors also refer to "deterministic models" and "ensemble models". This should be replaced with "deterministic forecasts" and "ensemble forecasts" throughout the text as both actually use the same model (UM).*
   **Reply 2:** We have replaced *"deterministic models" and "ensemble models"* with *"deterministic forecasts" and "ensemble forecasts"* throughout the text.
3. *Page 1, Line 9 removed "the" in this sentence - here we investigate extreme events (heatwaves)*
   **Reply 3:** We have removed "the" from the sentence. This change can be seen on page 1 and line 9.
4. *Page 1, Line 22 - replace "...prediction the extreme events" with "...prediction of extreme events"* **Reply 4:** We have replaced *"…prediction the extreme events" with "...prediction of extreme events"* in page 1, line 22.
5. *Page 1, Line 22 - I don't understand the sentence "Extreme Weather events comprehend non-linear interactions..."*
   **Reply 5:** The said sentence is replaced as follows in the first para in Introduction on page 1 lines 23-25.

   Severe weather events (thunderstorms, cloudburst, heatwaves and coldwaves etc) usually involve strong non-linear interactions ,often between small scale features in the atmosphere (Legg and Mylne, 2004 ). For example, development of deep convection and thunderstorms in the tropics.

6. *Page 1, Line 30 - simplify this sentence "Based on multiple perturbed initial conditions, ensemble approach samples the errors in the initial..." to "It is based on*
   **Reply 6:** At page 1, Line 32 has been simplified and added "It is" in the beginning of the sentence.
7. *Page 2, line 1 - remove the first reference to Sarkar et. al., 2009, as it is repetitive.*
   **Reply 7:** The reference 'Sarkar et. Al., 2009' has removed from the text on page 2 line 4.
8. *Page 2 line 2 - Replace "Met office" with "Met Office"* 2
   **Reply 8:** "Met office" replaced with "Met Office" on page 2 line 5.
9. *Page 2 line 14 - replace "0.85 0°C" with "0.85 °C"*
   **Reply 9:** In the text we have replaced "0.85 0°C" with "0.85 °C" in line 18 on page 2.
10. *Page 2 line 15 - don't understand how Molteni et. al. (1996) could be used as refrence for a warming trend covering 1880-2012!*
    **Reply 10:** We have removed *Molteni et. al. (1996)* from the text. It was inadvertently typed on page 2 line 19.
11. *Page 2 line 17 - assume that the annual mean temperature of 0.42 C per 100 years refers to the globally averaged temperatures. This should be made clear.*
    **Reply 11:** We have elaborated in the text and updated with the new reference "*Kothawale et. al. 2010*" in line 20 on page 2.
12. *Page 2 line 21 - this paragraph begins with a sentence "Another study…" but the reference at the end of the sentence is Arora et. al. (2009) which was the same study discussed in the previous paragraph.*

**Reply 12:** Same as reply 11.

13. *Page 2 Line 24 - not sure what "recently reiterated" means?*
    **Reply 13:** Same as reply 11

14. *Page 2 line 28 - "sales" should read "scales"*
    **Reply 14:** We have replaced sales with scales now in line 3 on page 3.

15. *Page 2 line 29 - the sentence "...using ensemble forecast forecasts both, deterministic and ensemble forecasting." is very convoluted, can I suggest "...using both ensemble and deterministic forecasts"*
    **Reply 15:** The sentence is modified as per the suggestion in place of the *"...using ensemble forecast forecasts both, deterministic and ensemble forecasting."* to *"...using both ensemble and deterministic forecasts"* in the text in line 4 on page 3.

16. *Page 3 line 9 - delete "and" in the following "...adopt and the most compatible score type"*
    **Reply 16:** We have deleted "and" from the sentence "...adopt and the most compatible score type" from the text in line 18 on page 3.

17. *Page 3 line 11 - this sentence is very repetitive.*
    **Reply 17:** We have removed the repetitive sentence from the revised manuscripts.

18. *Page 3 line 23 - remove "... which was 1°x1° resolution a few years earlier over Indian land area." As it is irrelevant for this study.*
    **Reply 18:** We have removed the sentence on page 4 in line 1 "... which was 1°x1° resolution a few years earlier over Indian land area." from the revised manuscripts on page 4 line 1.

19. *Page 3 Line 32 - replace "operational NCMRWF" with "operational at NCMRWF*
    **Reply 19:** In the text we have replaced "operational NCMRWF" with "operational at NCMRWF"

20. *Page 4 line 8 - replace "...MET Office" with "...Met Office MOGREPS system (Bowler et. al. 2008)" where reference is Bowler, N. E., Arribas, A., Mylne, K. R., Robertson, K. B. and Beare, S. E. (2008), The MOGREPS short-range ensemble prediction system. Q.J.R. Meteorol. Soc., 134: 703–722. doi:10.1002/qj.234*
    **Reply 20:** "...MET Office" replaced ( on page 4 line 25) with "...Met Office MOGREPS system (Bowler et. al. 2008)" and reference is Bowler, N. E., Arribas, A., Mylne, K. R., Robertson, K. B. and Beare, S. E. (2008), The MOGREPS short-range ensemble prediction system. Q.J.R. Meteorol. Soc., 134: 703–722. doi:10.1002/qj.234 added in the reference list on page 11 lines 27-28.

21. *Page 4 Line 14 - replace "Uncertainty in forecasting model..." with "Uncertainty in the forecasting model..."*
    **Reply 21:** The sentence "Uncertainty in forecasting model..." replaced with "Uncertainty in the forecasting model..." in the revised manuscripts the change can be seen at page 4, line 32.

22. *Page 4 line 16 - Remove this line as it is repetitive (see line 4-5 on this page which says the same thing)*
    **Reply 22:** We have removed the line, "In this study, the forecast data is interpolated to match the grid and resolution of observations i.e. 0.5°x0.5° .for verification.", from the text on page 5 lines 1-2.

23. *Page 5 line 8 - Heidke skill score mentioned but not defined or used. Remove this reference?*
    **Reply 23:** Heidke skill (HK) score is used at page 5, from line 16-20 and at page 9,line 15.

24. *Page 5 line 26 - replace "...efficiency" with "...capability"?*
    **Reply 24:** The word "...efficiency" replaced with "...capability" at page 6, line 16.

25. *Page 6 line 9 - replace "... the figures (Fig. 5) and (Fig. 4)." with "Fig. 5 and Fig. 4."*
    **Reply 25:** The figures (Fig. 5) and (Fig. 4).", replaced with "Fig. 5 and Fig. 4."

26. *Page 6 line 11 - use "The deterministic forecast..."*
    **Reply 26:** The sentence started with the *"The deterministic forecast..."* in the text.

27. *Page 6 lines 11-12 Replace "...any variation in either of the considered days and illustrates quasi-stationary characteristics of the deterministic forecast from Day-1 through Day-10 forecast" with "... any variation in either of the days and illustrates quasi-stationary characteristics from Day-1 through Day-10"*
    **Reply 27:** The text is modified as per the suggestion on.
28. *Page 6 line 13 - Remove "...and vary in not so distinctive fashion".*
    **Reply 28:** A part of sentence *"...and vary in not so distinctive fashion"* is removed from the text.
29. *Page 6 line 15 - "Fig. (2.)" should read "Fig.2"*
    **Reply 29:** "Fig. (2.)" replace with "Fig.2"
30. *Page 6 line 15 - Remove "..(Tmax).."*
    **Reply 30:** Tmax removed from the text.
31. *Page 8 line 7 - Fig 10. should read Fig 9.*
    **Reply 31:** The figure " Fig 10" is replace with "Fig 9" in the text.
32. *Page 8 line 18 - Fig. 9 should read Fig. 10.*
    **Reply 32:** The figure " Fig 9" is replace with "Fig 10" in the text.
33. *Page 8 line 21 - missing end parentheses ")"*
    **Reply 33:** The missing end parentheses ")" is inserted in the text.
34. *Page 8 line 23 - missing figure number.*
    **Reply 34:** The missing figure number is inserted in the text.

**Figures and tables**

1. *Figure 2 and 3 - the colour bar is labelled °C When the quantity is a count.*
   **Reply 1:** The colour bar label °C is removed from the figure 2.
2. *Table 2 title - "Causalities" should read "Casualties".*
   **Reply 2:** The word "Causalties" replaced by the "Casualties" in the table title.

[revised manuscript text omitted]

---

## Editor Decision (ED1)

**Verification of Pre-Monsoon Temperature Forecasts over India during 2016 with focus on Heatwave Prediction**

Harvir Singh, Kopal Arora, Raghvendra Ashrit, and EN Rajagopal

Ministry of Earth Sciences, National Centre for Medium Range Weather Forecasting, Noida, 201309, India

5   *Correspondence to*: Harvir Singh (harviriitkgp@gmail.com)

**Abstract.**   The operational medium-range weather forecasting based on Numerical Weather Prediction (NWP) models are complemented by the forecast products based on Ensemble Prediction Systems (EPS).  This change has been recognized as an essentially useful tool for the medium range forecasting and is now finding its place in forecasting the extreme events. Here we investigate extreme events (Heatwaves) using a high-resolution numerical weather prediction and its ensemble

10   forecast in union with the classical statistical scores to serve the verification purposes. With the advent of climate change related studies in the recent past, the rising extreme events and their plausible socio-economic effects have encouraged the need for forecasting and verification of extremes. Applying the traditional verification scores and the associated methods on both, deterministic and the ensemble forecast, we attempted to examine the performance of the ensemble based approach as compared to the traditional deterministic method.  The results indicate

15   towards an appreciable competence of the ensemble forecasting detecting extreme events as compared to deterministic forecast. Locations of the events are also better captured by the ensemble forecast.  Further, it is found that the EPS smoothes down the unexpectedly soaring signals, which thereby reduce the false alarms and thus prove to be more reliable than the deterministic forecast.

**1.   Introduction**

20   Reliable weather forecasting plays a pivotal role in our everyday activities.  Over the years NWP systems have been employed to serve the purpose. While the NWP models have demonstrated an improved forecasting capability in general, they still have a challenge in the accurate prediction of severe weather/extreme events.  Severe weather events (thunderstorms, cloudburst, heatwaves and coldwaves etc) usually involve strong non-linear interactions ,often between small scale features in the atmosphere (Legg and Mylne, 2004 ). For example, development of deep convection and

25   thunderstorms in the tropics.  These small-scale interactions are difficult to predict accurately (Meehl et al., 2001) and a small deviation in these could lead to completely different results, as a result of the forecast evolution process (Lorenz, 1969). The inherent uncertainty in the weather and climate forecasts can be well handled by employing ensemble based forecasting (Buizza et al., 2005).  The EPS (Mureau et al., 1993, Toth and Kalnay, 1997, Molteni et al., 1996) were first introduced in the 1990s in an effort to quantify the uncertainty caused by the synoptic scale baroclinic instabilities in the

30   medium range weather forecasting (Legg and Mylne,2004). Ensemble forecasting has emerged as the practical way of estimating the forecast uncertainty and making probabilistic forecasts. It is based on multiple perturbed initial conditions, ensemble approach samples the errors in the initial conditions to estimate the forecast uncertainty (spread in member

**Summary of Comments on Microsoft Word - Temp_verification-Revised-05Apr2017**

**Page: 1**
* * *
**Number: 1**     Author: Bruce     Subject: Highlight Date: 06-Apr-17 09:44:59

Join up with first part of abstract (no paragraph break).
* * *
**Number: 2**     Author: Bruce     Subject: Highlight Date: 06-Apr-17 09:26:42

"etc" should be "etc." and always have a comma before it. Please correct everywhere.
* * *
**Number: 3**     Author: Bruce     Subject: Highlight Date: 06-Apr-17 09:28:21

Please put oldest to newest. So 1993, 1996, 1997. Check entire manuscript.

Please go to the NHESS WORD TEMPLATE FOR AUTHORS and AUTHOR GUIDELINES FOR SUBMISSION as I believe that these need to be separated by ;

forecasts). The skill of the ensemble forecast shows marked improvement over the deterministic forecast when comparing the ensemble mean to deterministic forecast after a short lead time

The new EPS at the NCMRWF is now running for operational purposes. This global medium-range weather forecasting system has been adopted from the UK Met Office (Sarkar et al., 2016). Generally, the model and the ensemble forecast applications in addition to their verifications are used for prevalent events with a limited focus on the rare extreme weather events. It would be for the first time that the EPS technique has been employed from this model output for the extreme events over India to study the heatwave events. The heatwave is considered if maximum temperature of a station reaches at least 40°C or more for Plains and at least 30°C or more for Hilly regions. Based on departure from normal, a station is declared to have heatwave conditions if departure from normal is 4.5°C to 6.4°C and severe heatwave if the departure from normal is >6.4°C. In terms of the actual maximum temperature, a station is under heatwave when actual maximum temperature ≥ 45°C and severe heatwave when the maximum temperature is >47°C. There has been increasing interest in predicting such extremes, the heatwave and cold wave events in India due to the associated loss of life. An increasing number of extreme temperature events over India were documented by a few recent studies (Qin et al., 2013). [1] study conducted over the Indian sub-continent between 1969 and 1999 indicated more frequent cold and heatwave events over the Indo-Gangetic plains of India. [2] 6 heatwave events and 2-3 cold wave events are reported to occur every year in the Northern parts of the country. The global temperatures have exhibited a warming trend of about 0.85°C due to anthropogenic activities between 1880 and 2012. Similar trends were also observed in India with the annual air surface temperature rise during 20$^{th}$ century. This is evident from the detailed study presented in Kothawale et al (2010) based on the data from 1901-2007.

[3] he Indian mean maximum and minimum annual temperatures have significantly increased by 0.51, 0.71 and 0.27$^{\circ}$C per 100 years respectively, during 1901-2007. However, an accelerated warming was observed during 1971-2007, mainly due to the last decade 1998-2007. [4] he study highlights that the mean temperature during the pre-monsoon season (March-May) shows an increasing trend of 0.42°C per 100 years. On the other hand, a recently reiterated IPCC report (2013) notified an "unequivocal" proof of the increasing warming trend, globally which could be associated with the variations in the climate system. This indicates a need to comprehend the heatwave events on weather and climatic scales. [5] hile there is an extensive literature discussing the heatwave events and their trends on the climatic scales, however, the literature is rather limited (especially over India) focusing such events on monthly scales. This paper thus tries to fill in the gap and attempt to demonstrate the capability and strength of predicting such events using both ensemble and deterministic forecast. This research investigates the most recent heatwave events during the summer months March, April & May (MAM) 2016 in India. This investigation considers two case studies to demonstrate the strength and weaknesses of the EPS approach in predicting such extreme events.

With these factors in mind, we can say that temperature (Minimum and Maximum both), forms a vital component of weather and climatic studies which are becoming increasingly important and challenging. Reliable projections of such changes in
* * *
**Number: 1        Author: Bruce        Subject: Highlight Date: 06-Apr-17 09:31:02**

Important for entire manuscript: Where are these facts from? Check EVERY sentence, and ensure it is clear where all facts/information are from. So I look below, in next paragraph, and it is unclear.
* * *
**Number: 2        Author: Bruce        Subject: Highlight Date: 06-Apr-17 09:30:08**

Avoid starting a sentence with a number. PLEASE CHECK EVERYWHERE IN MANUSCRIPT.

You could do "There are reported to occur every your in the Northern parts of the country 5-6 heatwave events and 2-3 cold wave events.

IMPORTANT: Make sure that it is clear where these facts comes from. Not clear.
* * *
**Number: 3        Author: Bruce        Subject: Highlight Date: 06-Apr-17 09:31:34**

Where are these facts from?
* * *
**Number: 4        Author: Bruce        Subject: Highlight Date: 06-Apr-17 09:31:22**

What study?
* * *
**Number: 5        Author: Bruce        Subject: Highlight Date: 06-Apr-17 09:32:59**

Give half a dozen examples please.

....climatic scales (e.g., ****, ****, ****, ****). However, the literature is...

Tennant, W.J., Shutts, G.J., Arribas, A. and Thompson, S. A.: Using a Stochastic Kinetic Energy Backscatter Scheme to Improve MOGREPS Ensemble forecast Skill. *Mon. Weather Rev.*, **139**, 1190-1206, 2010.

Toth, Z., and E. Kalnay: Ensemble forecasting at NCEP and the breeding method. Mon. Wea. Rev, **125,** 3297–3319, 1997.

[Figure]

Figure 1. Frequency distribution of observed, and forecast (NCUM and NEPS) (a) *Tmax* (°C) and (b) *Tmin* (2C)  over India
10   during March-May 2016.

**Page: 13**

[Figure]

For every figure, make sure it is clear WHERE the data has come from (Data from *****). This should then be in the reference list.

For x- and y-axis here and ALL other figures, please put in italic the variables, so they are the same as in text.

Make a decision between max and min being sub-script or not sub-script, you have both (sub-script in graphs, non-subscript in text).

Figure 2. Spatial distribution of observed and NCUM forecasts number of days with *Tmax* ≥ 40°C during the period of March to May 2016

**Page: 14**

[Figure]

Legend here and ALL figures, make sure that units are indicated.

Please ensure that any acronyms or variables are clear and self-standing in figure captions, so reader does not have to go to text to figure them out.

Figure 4. Mean Sea Level Pressure (MSLP) shaded and winds at 700 hPa showing heat low (a) Analysis of 20160410 (b) Day 3 forecast valid for 20160410 (c) Analysis of 20160521 (d) Day 3 forecast valid for 20160521

16

Make sure that your final version of this figure is higher resolution.

Please ensure that legend has units.

[Figure]

Figure 9. Box plots for HK scores for different temperature ranges (*Tmax*) NCUM and NEPS form March to May 2016

[Figure]

Figure 10. Box plots for Equitable Threat Score (ETS) for NCUM and NEPS form March to May 2016

21

**T** Number: 1     Author: Bruce     Subject: Highlight Date: 06-Apr-17 09:40:30

There are many kinds of box-plots. Please indicate in figure caption text, what the whiskers represent (could be max, min, could be 95% 5%), what the horizontal line represents (mode?), and what the plus represents.

Do not assume the reader will go to the text.

**T** Number: 2     Author: Bruce     Subject: Highlight Date: 06-Apr-17 09:41:12

Do another read through for typos like this. "Should be "from" not "form". Give this to a student to read carefully--copy-editors can only pick up so much.

**Table 2.** asualities reported during MAM-2016 due to prevailing heatwaves over India

| Month | State/ Region | No. of loss of lives | Total |
|---|---|---|---|
| March | Maharashtra | 1 | 2 |
| | Kerala | 1 | |
| April | Odisha | **88** | 220 |
| | Telangana | 79 | |
| | AP | 40 | |
| | Maharashtra | 9 | |
| | Karnataka | 1 | |
| | Tamil Nadu | 1 | |
| May | Telangana | **200** | 273 |
| | Gujrat | 39 | |
| | Maharashtra | 34 | |

Table 3. Monthly *Tmax* > 40°C scores for CUM and NEPS forecast with IMD observed temperature

| Month | Score | NCUM | | | | | NEPS | | | | |
|---|---|---|---|---|---|---|---|---|---|---|---|
| | | Day 1 | Day 3 | Day 5 | Day 7 | Day 9 | Day 1 | Day 3 | Day 5 | Day 7 | Day 9 |
| MAR | POD | 0.25 | 0.23 | 0.27 | 0.30 | 0.28 | 0.23 | 0.20 | 0.22 | 0.24 | 0.22 |
| | FAR | 0.81 | 0.71 | 0.75 | 0.75 | 0.79 | 0.49 | 0.54 | 0.53 | 0.53 | 0.43 |
| | ETS | 0.09 | 0.09 | 0.09 | 0.08 | 0.08 | 0.10 | 0.09 | 0.10 | 0.11 | 0.11 |
| | HK | 0.22 | 0.21 | 0.24 | 0.27 | 0.25 | 0.21 | 0.18 | 0.21 | 0.23 | 0.21 |
| | SEDI | 0.33 | 0.32 | 0.36 | 0.38 | 0.36 | 0.31 | 0.30 | 0.34 | 0.34 | 0.33 |
| APR | POD | 0.39 | 0.39 | 0.38 | 0.36 | 0.36 | 0.43 | 0.43 | 0.41 | 0.42 | - |
| | FAR | 0.66 | 0.65 | 0.66 | 0.66 | 0.66 | 0.62 | 0.61 | 0.62 | 0.61 | 0.62 |
| | ETS | 0.16 | 0.16 | 0.15 | 0.15 | 0.15 | 0.19 | 0.19 | 0.19 | 0.19 | 0.19 |
| | HK | 0.30 | 0.29 | 0.28 | 0.27 | 0.26 | 0.34 | 0.34 | 0.34 | 0.33 | 0.33 |
| | SEDI | 0.46 | 0.45 | 0.45 | 0.43 | 0.42 | 0.51 | 0.51 | 0.52 | 0.51 | 0.50 |
| MAY | POD | 0.30 | 0.30 | 0.28 | 0.26 | 0.24 | 0.32 | 0.34 | 0.31 | 0.31 | 0.27 |
| | FAR | 0.70 | 0.71 | 0.72 | 0.74 | 0.75 | 0.67 | 0.69 | 0.70 | 0.71 | 0.75 |
| | ETS | 0.12 | 0.11 | 0.11 | 0.10 | 0.09 | 0.14 | 0.14 | 0.13 | 0.12 | 0.10 |
| | HK | 0.22 | 0.22 | 0.21 | 0.19 | 0.17 | 0.25 | 0.26 | 0.24 | 0.23 | 0.19 |
| | SEDI | 0.39 | 0.38 | 0.36 | 0.33 | 0.30 | 0.43 | 0.43 | 0.40 | 0.39 | 0.33 |

Number: 1     Author: Bruce     Subject: Highlight Date: 06-Apr-17 09:44:02

Give citation to data, what MAM means, and POPULATION of these regions (otherwise hard to put into perspective of what the loss of lives means if not normalized.

Number: 2     Author: Bruce     Subject: Highlight Date: 06-Apr-17 09:42:37

In Table headers and figure captions, these should be self-standing. So reader should not have to go to the text to figure out what they mean. Give what the acronyms mean. Tell us where the data is from. Tell us the period of the data.

---

## Author Response (AR2)

Title: Verification of Pre-Monsoon Temperature Forecasts over India during 2016 with focus on Heat Wave Prediction

NHESS-2016

We are thankful to the editor and the reviewer for their helpful suggestions which have helped us to improve the quality of the paper to a great extent. We have tried to incorporate as many of their suggestions as possible.

*How much of the skill in predicting the heatwaves comes from persisting a heatwave already present in the initial conditions? How does the model perform when the heatwave evolves within the forecast range (e.g. Beyond days 2-3).*

Reply 1: Extreme events like heat waves are rare in nature and here we provided a general view of the two particular heat wave events (11 April & 21 May). From our experience as well as the forecast for the post heat wave event days, we can state that the skill of predicting an event with the initial conditions of *no indication* of severity is comparatively *lower* than when the signature is present in the initial conditions.
Even before the event, there is some signature of it as can be seen in the figure (). The overall prediction of warm conditions is nicely predicted but at closer lead times, the events are better predicted. Same can be seen in the box and whisker plots for ETS (and rest of the score plots as well). For instance, the skill of NEPS does not fall drastically from Day-2 to Day-7 and thus depicts a reasonable skill. So, overall the NEPS specifically, has a good skill in predicting the extreme event and is relatively robust. Now we have included this reply in the manuscript in the end of conclusions.

"...converg towards.." should read "...converge towards..."

Reply 2: We have corrected the phrase in the manuscript, "...converg towards.." to"...converge towards..."

Title: **Verification of Pre-Monsoon Temperature Forecasts over India during 2016 with focus on Heat Wave**

**Prediction**

**NHESS-2016**

**We are thankful to the editor and the reviewer for their helpful suggestions which have helped us to improve the quality of the paper to a great extent. We have tried to incorporate as many of their suggestions as possible.**

Reviewer 2, Comment 1 - The authors have addressed my comment in their reply but not incorporated any discussion into the paper. Just a sentence along the lines of their reply would be useful.

**Reply Reviewer 2, comment 1:** We have elabaorated the reviewer #2 comment 1 in the revised Manuscrits and added a paragraph from line 13 to 21 at page 10.

Section 3 - "...converg towards.." should read "...converge towards..."

**Reply Section 3 : Now we have  corrected the spelling of converge in manuscripts at page 6, line 11.**

---

## Editor Decision (ED2)

**Verification of Pre-Monsoon Temperature Forecasts over India during 2016 with focus on Heat Wave Prediction**

Harvir Singh, Kopal Arora, Raghavendra Ashrit, and EN Rajagopal

Ministry of Earth Sciences, National Centre for Medium Range Weather Forecasting, Noida, 201309, India

5  *Correspondence to*: Harvir Singh (harviriitkgp@gmail.com)

**Abstract.** The operational medium-range weather forecasting based on Numerical Weather Prediction (NWP) models are complemented by the forecast products based on Ensemble Prediction Systems (EPS). This change has been recognized as an essentially useful tool for the medium range forecasting and is now finding its place in forecasting the extreme events. Here we investigate extreme events (heat waves) using a high-resolution numerical weather prediction and its ensemble

10  forecast in union with the classical statistical scores to serve the verification purposes. With the advent of climate change related studies in the recent past, the rising extreme events and their plausible socio-economic effects have encouraged the need for forecasting and verification of extremes. Applying the traditional verification scores and the associated methods on both, deterministic and the ensemble forecast, we attempted to examine the performance of the ensemble based approach as compared to the traditional deterministic method. The results indicate towards an appreciable competence of the ensemble

15  forecasting detecting extreme events as compared to deterministic forecast. Locations of the events are also better captured by the ensemble forecast. Further, it is found that the EPS smoothes down the unexpectedly soaring signals, which thereby reduce the false alarms and thus prove to be more reliable than the deterministic forecast.

**1. Introduction**

Reliable weather forecasting plays a pivotal role in our everyday activities. Over the years NWP systems have been

20  employed to serve the purpose. While the NWP models have demonstrated an improved forecasting capability in general, they still have a challenge in the accurate prediction of severe weather/extreme events. Severe weather events (thunderstorms, cloudburst, heat waves, and cold waves, etc.) usually involve strong non-linear interactions, often between small scale features in the atmosphere (Legg and Mylne, 2004 ). For example, development of deep convection and thunderstorms in the tropics. These small-scale interactions are difficult to predict accurately (Meehl et al., 2001) and a

25  small deviation in these could lead to completely different results, as a result of the forecast evolution process (Lorenz, 1969). The inherent uncertainty in the weather and climate forecasts can be well handled by employing ensemble based forecasting (Buizza et al., 2005). The EPS (Mureau et al., 1993; Molteni et al., 1996; Toth and Kalnay, 1997) were first introduced in the 1990s in an effort to quantify the uncertainty caused by the synoptic scale baroclinic instabilities in the medium range weather forecasting (Legg and Mylne 2004). Ensemble forecasting has emerged as the practical way of

30  estimating the forecast uncertainty and making probabilistic forecasts. It is based on multiple perturbed initial conditions, ensemble approach samples the errors in the initial conditions to estimate the forecast uncertainty (spread in member

**Summary of Comments on Microsoft Word - nhess-2016-264-manuscript-version7**

**Page: 1**

Number: 1     Author: Bruce     Subject: Inserted Text     Date: 02-May-17 17:40:02
Numerical Weather Prediction (NWP)

Number: 2     Author: Bruce     Subject: Inserted Text     Date: 02-May-17 17:40:33
Ensemble Prediction Systems (EPS)

Number: 3     Author: Bruce     Subject: Highlight Date: 02-May-17 17:42:05
Needs a reference or references. This is a statement and needs support by the literature.

Number: 4     Author: Bruce     Subject: Highlight Date: 02-May-17 17:41:14
Insert space

[revised manuscript text omitted]

---

## Author Response (AR3)

Replies to comments and suggestions: Revised 5[th] Apr 2017-04-14

**Page 1:**

Page 1: Number 1:  As suggested the para break is removed. The chage can be seen on line 10 page 1 in the revised manuscript.

Page 1: Number 2: This typographical error is corrected in the entire manuscript and can be tracked in the manuscript at two places only.

Page 1: Number 3: The order of the cited literature is now in chronological order in the manuscript. This can be tracked.

**Page 2:**

Page 2: Number 1:The text is appropriately modified in the revised manuscript. Several references have been added on page 2: line 10 and 20. Additionally redundant statements in the earlier version of the manuscript have been done away with which can also be tracked on Page 2

Page 2: Number 2: That was an un necessary statement and had not credible support and hence has been avoided in the revised manuscript.

Page 2: Number 3: Suitable references have been included.

Page 2: Number 4: The text is the revised manuscript is modified to indicate the referred study. This appears on line 27 page 2.

Page 2: Number 5: This was another unnecessary statement and has been avoided in the revised manuscript.

**Page 13:**

Page 13: Number 1: The suggested changes are reflected in the new improved figure and caption on page 14 o fthe revised manuscript.

Page 13: Number 2: The x- axis now has the symbol for the degree C. All the text and diagrams now feature 'max'and 'min' in Tmax and Tmin with out subscript.

**Page 14:**

Page 14: Number 1: The figure shows the counts and is indicated in the caption. Hence the legend is undisturbed in the revised manuscript. However, as suggested , the information on data source etc is included.

Page 14: Number 2: The acronyms of IMD, NCUM and NEPS have been expanded in each of the figure captions.  It must however be noted that NCUM stand for NCMRWF Unified Model. Further expansion of NCMRWF (name of the NWP centre in India) is avoided to keep the captions brief.

**Page 16**

Page 16: Number 1: The improved graphic of Figure 4 now show the units alongside the legend which can be seen on Page 17 in the revised manuscript.

**Page 21:**

Page 16: Number 1: The Box-Whisker plots now show the legend describing the components of the box-plot. These now appear on pages 22-24 in the revised manuscript.

Page 16: Number 2: We thanks for bringing this to our notice. As suggested we have eliminated such errors in the revised manuscript.

**Page 25:**

Page25: Number 1: Table 2 now included the detailed information on the source of the data in the bottom of the table on Page 26 in the revised manuscript.

Page25: Number 2: We have made all the table header and all figure captions self sufficient to the extent possible.

---

## Author Response (AR4)

**Reply to Comments and suggestions**: **Revised on 1st May 2017**

(a) Everywhere, double check for typos. I saw some, copy-editor will pick up on others, but now is the time for you to pick up on these, BEFORE it goes to copy-editing. So missing spaces. Or, "form" instead of "from". Please read the text carefully.

*Response (a) :As pointed out there were several such typos and we have taken care to eliminate all such typographical errors. We have eliminated missing spaces at several places. Additionally we have made consistent use of term 'heat wave' (and not 'heatwave') in the revised text. Appropriate use of articles is also enhanced in the revised text.*

*The revised manuscript is uploaded in .pdf form under Manuscript(pdf) in clean form.*

(b) Make sure ALL figures are of high-enough resolution that they will appear well in the final version online. Some are very low resolution.[better, but still needs work]

*Response (b) :In the previous revision we ensured that all the figures have high resolution (300dpi) and same is maintained in the present version.*

(c) Throughout your text, would you please ensure that it is very clear where facts and information are from. In other words--an in-text citation (reference) which could be grey literature or peer-review literature, or perhaps information from a figure. There are many places in your text this is not clear. So you state facts/information, but not clear where these are from.

*Response (c) :We took care of this in the previous (6th version) revision of the manuscript. Accordingly we have included several references.*

Other:

(d) Please put abbreviations in alphabetical order by the abbreviation.

*Response (d) :This is also done in the table on page 25.*

[revised manuscript text omitted]

Comment [G21]: Deleted:p ok a toll of over 200 lives foerlier

Comment [G19]: Inserted: -

Comment [G20]: Inserted: of

Comment [G22]: Inserted: ,

**4.3.1 Case-I Heat waves on 11th April 2016**

As per the IMD reports (Climate Diagnostics Bulletin of India, April 2016), the heat wave conditions prevailed over parts of central peninsular and east India during the second week of the April. It took a toll of over 200 lives (Table-2) from central and peninsular India during the April month. Observed and forecast *Tmax* valid for 11th April 2016 is shown for NCUM (Figure 5) and NEPS (Figure 6). The spatial distributions of *Tmax* show prevailing heat-waves over Odisha, AP, Telangana, and some parts of Maharashtra on 11th April 2016. The observation shows more than 40°C spread over east UP, Bihar, West Bengal, east MP, Jharkhand, Chhattisgarh, Odisha, Maharashtra and Some parts of Karnataka and Tamil Nadu. In the NCUM forecast, on other hand showing marginally wider regions up to day-9 due to a warm bias in the model and on the contrary NEPS forecasts also showing ×40°C wider regions up to day-9 but marginally less than the NCUM forecasts. Apart from the warm bias, both the model forecast is showing cold bias in north-northeast of J&K. Hence theNEPS is better in predicting the extremes of heat waves up to Day-9 then the NCUM.

**4.3.2 Case-II Heat waves on 21st May 2016**

The severe heat wave conditions developed and intensified over parts of northwest India entire third week of May-2016 and persisted over parts of central and north peninsular India some stations of West Rajasthan temperature observed × 50°C viz. Barmer, Bikaner, Ganganagar, Jaisalmer & Jodhpur and observed severe heat wave conditions for 4 to 5 days in succession from 17th to 21st May-2016. The spatial distributions of NCUM & NEPS forecast *Tmax* with of observed IMD *Tmax* prevailing heat-waves over Rajasthan, MP, UP, Delhi, Haryana, Punjab and some parts of Maharashtra on 21st May 2016 is shown in Figure 7 & 8. Both the models deterministic and ensemble able to predict the extreme temperature (*Tmax* > 48°C) over west Rajasthan up day-3 only. However, the NCUM is predicting more wide-spreading *Tmax* > 46°C, over Rajasthan, MP, UP, Delhi, Haryana, Punjab and parts of Maharashtra all days forecast.

The H-K scores of the maximum temperature (*Tmax*) between the range 30-42 °C, constructed as box and whiskers for both NCUM and NEPS, indicate towards better performance of the ensemble based forecast as compared to the deterministic one. Interestingly, the forecast score does not fade away with the lead time contrary to the expectation. This depicts that the NEPS performs better and its prediction skill remains quasi-constant throughout the lead time of 10 days (Figure 9).

Similar observations can be made from the ETS plots (Figure 10).The most obvious finding to emerge from the box and whiskers plots of the ETS scores is the better performance of the ensemble based forecast (NEPS) than that of the deterministic forecast (NCUM). This result is consistent with the earlier documented findings. At all the *Tmax* thresholds (between 30 and 42°C), NEPS mean stands above the NCUM mean. The same observation holds during all the illustrated forecasts (Day1, 3, 5, 7, and 9). The scores falling under the 25% in the case of the ensemble based forecast are either similar

**Comment [G26]:** Deleted:s

**Comment [G23]:** Inserted: ,

**Comment [G24]:** Inserted: a

**Comment [G25]:** Inserted: D

**Comment [G27]:** Deleted:h

**Comment [G28]:** Deleted:s

**Comment [G29]:** Deleted:r

**Comment [G30]:** Deleted:d

[revised manuscript text omitted]

---

## Author Response (AR5)

**Reply to Comments and suggestions**: **Revised on 9th May 2017**

**Page1: Number 1**: In the abstract and in the introduction Numerical Weather Prediction (NWP) is used. Subsequently in the text abbreviation NWP is used.

**Page1: Number 2**: In the abstract and in the introduction Numerical Ensemble Prediction System (EPS) is used. Subsequently in the text abbreviation EPS is used.

**Page1: Number 3**: The reference of Buizza et al., 2005 and Ashrit et al., 2013 are given in the revised manuscript. The manuscript is appropriately modified with correct spacing and use of articles at several places. Additional care is also taken to check spelling and grammar.

**Page2: Number 1**: The statement is now supported with reference of Buizza et al., 2005 in the revised manuscript.

**Page2: Number 2**: The text is modified as suggested and we have indicated (November 2015) the starting month and year of the Ensemble Model.

**Page2: Number 3**: The text in the revised manuscript is suitably modified as

 õ*interest among the general public, media and local administration in predicting such extremes*,ö

**Page2: Number 4**: The sentence is modified to clarify as below-

õ*Current paper attempts to demonstrate the capability and strength of predicting such events using both ensemble and deterministic models. Present study investigates the heat wave events during the summer months March, April & May (MAM) 2016 in India*ö

**Page3: Number 1,2 and 4**: The sentence is eliminated in the revised manuscript since it was redundant.

**Page3: Number 3:** The reference of WMO 2008 is included in the revised manuscript.

**Page3: Number 5 and 6:** The contractions have been eliminated and use of words like ÷*recently*ø is avoided in the manuscript.

**Page3: Number 7,8 and 9:** All missing spaces have been removed, typographical error are corrected in usage of ÷ø, ÷øand ÷ø

**Page4: Number 1:** The usage of õet al.ö -is corrected in the manuscript to make it consistent with other papers published in the NHESS  and as per the instructions to the authors.

**Page4: Number 2:** The most suitable reference of Wilks 2011 in the beginning of section 2.4. Thanks for the suggestion.

**Page5: Number1, 2 and 3:** Names of all the scores have been expanded in their first use along with the abbreviations in the bracket. In the subsequent appearance in the text abbreviations are used.

**Page6: Number 1:** First couple of general sentences are eliminated in the revised manuscript and references are not needed.

**Page6: Number 2:** The style of referring to a figure in the middle of sentence and in the beginning of the sentence is followed as per the Guidelines to authors and as in other papers of NHESS.

**Page6: Number 3&4:** In the revised manuscript the two suggested changes have been incorporated. The forecast lead times are referred to as õday 2ö and month and year are referred to as õMarch 2016ö.

**Page6: Number 5**: The official document Climate Diagnostics Bulletin of India published by the India Meteorological Department is cited and is included in the references.

**Page7: Number 1&2:** The official document Climate Diagnostics Bulletin of India published by the India Meteorological Department is cited and is included in the references.

**Page7: Number 3:** The revised text now consistently has the forecast lead time referred to as õday #ö.

**Page7: Number 4:** The use of initials while citing is carefully avoided in the revised text.

**Page22: Number 1:** The figures in the revised manuscript are significantly higher in resolution (600 dpi). Each of the plots has been re-done to clearly show the fonts in all labels.

**Page25: Number 1&2:** The abbreviations and their descriptions used in the manuscript are listed in Table 1 in alphabetical order. The manuscript is carefully edited to prevent unnecessary underlining.

**Additionally**

All the figure captions in all the figures have now been significantly modified to explain the contents and inference.

[revised manuscript text omitted]

J.VG., Vincent, G., L., Stephenson, L., D., Burn, D.,J., E. Aguilar, J. E.,M. Brunet, M.,. Taylor, M.,. New, M.,P. Zhai, P.,M. Rusticucci, M.,J. L. Vazquez-Aguirre, J.L.,: Global observed changes in daily climate extremes of temperature and precipitation, Journal of Geophysical Research - Atmospheres, Vol. 111 (D05109), doi:10.1029/2005JD006290, 2006.

5 Ashrit, R., Iyengar, G. R., Sankar, S., Ashish, A., Dube, A., Dutta, S. K., Prasad, V. S., Rajagopal, E. N., and Basu S.: Performance of Global Ensemble Forecast System (GEFS) During Monsoon 2012. Research Report, NMRF/RR/01/2013, 25, 2013

Bishop, C. H., Etherton, B. J.,. and Majumdar, S. J.: Adaptive sampling with the ensemble transform Kalman filter. Part I: 10 Theoretical aspects,. Mon. Wea. Rev, **129,** 420‒436, 2001.

Bowler, N. E., Arribas, A., Mylne, K. R., Robertson, K. B. and Beare, S. E. ;(2008), The MOGREPS short-range ensemble prediction system,. Q.J.R. Meteorol. Soc., 134,: 703‒722. doi:10.1002/qj.234, 2008.

15 Buizza, R., Houtekamer, P.L., Toth, Z., Pellerin, G., Wei, M., Yuejian, Z.: A comparison of the ECMWF, MSC and NCEP global ensemble prediction systems,. Mon. Weather Rev. **133**,: 1076‒1097, 2005.

Caesar, J., Alexandar, L. and Vose, R.: Large-scale changes in observed daily maximum and minimum temperatures: Creation and analysis of a new gridded data set,. J. Geophys. Res., **111**, D05101, 2006.

20 Climate Diagnostics BulletinChange 2013: The Physical Science Basis, Working Group I Contribution to the Fifth Assessment Report of Indiathe Intergovernmental Panel on Climate Change (IPCC)

25 Climate Diagnostics Bulletin of India, April 2016, Near Real - Time Analyses National Climate Centre, Pune, April 2016.

Climate Diagnostics Bulletin of India, Near Real - Time Analyses National Climate Centre, Pune, March 2016, Near Real - Time Analyses National Climate Centre, Pune.

[revised manuscript text omitted]

---

## Author Response (AR6)

**Reply to Comments and suggestions**: **Revised on 4[th] August 2017**

**Page1:** The formatting of title, authors detail and their affiliation are modified as per the Copernicus.

**Page5:** The mathematical equations also modified as per the Copernicus.

Apart from these modifications all headings, references entire manuscript is formatted closely with the Copernicus.

Field Code Changed

[revised manuscript text omitted]

Field Code Changed

**Table 1: List of Abbreviations**

| | |
|---|---|
| **AP** | **A**ndhra **P**radesh |
| **EPS** | **E**nsemble **P**rediction **S**ystems |
| **ETKF** | **E**nsemble **T**ransform **K**alman **F**ilter |
| **ETS** | **E**quitable **T**hreat **S**core |
| **FAR** | **F**alse **a**larm **r**atio |
| **HK** | **H**anssen **and K**uipers |
| **IMD** | **I**ndian **M**eteorological **D**epartment |
| **J&K** | **Jammu & Kashmir** |
| **MAM** | **M**arch, April and **M**ay |
| **MOGREPS** | Met Office Global and Regional Ensemble Prediction System |
| **MP** | **M**adhya **P**radesh |
| **MSE** | **Mean Square Error** |
| **MSLP** | **Mean Sea Level Pressure** |
| **NCMRWF** | **N**ational **C**entre **for M**edium **R**ange **W**eather **F**orecasting |
| **NCUM** | **NCMRWF** U**nified M**odel |
| **NDC** | **N**ational **D**ata **C**entre |
| **NEPS** | **NCMRWF E**nsemble **P**rediction **S**ystem |
| **NWP** | **N**umerical **W**eather **P**rediction |
| **OPS** | **Observation Processing System** |
| **POD** | **P**robability **O**f **D**etection |
| **RMSE** | **Root Mean Square Error** |
| **SEDI** | **S**ymmetric **E**xtremal **D**ependence Index |
| ***Tmax*** | **M**aximum **Temperature** |
| ***Tmin*** | **M**inimum **Temperature** |
| **UK** | **United Kingdome** |
| **UM** | **Unified Model** |
| **UP** | **U**ttar **P**radesh |
| **WMO** | **World Meteorological Organization** |

**Table 2. : Casualties reported during March to May 2016 due to prevailing heat waves over India**

| Month | State/ Region | No. of loss of lives | Total |
|---|---|---|---|
| March | Maharashtra | 1 | 2 |
| | Kerala | 1 | |
| April | Odisha | **88** | 220 |
| | Telangana | 79 | |
| | AP | 40 | |
| | Maharashtra | 9 | |
| | Karnataka | 1 | |
| | Tamil Nadu | 1 | |
| May | Telangana | **200** | 273 |
| | Gujrat | 39 | |
| | Maharashtra | 34 | |

*(Data: Climate Diagnostic Bulletin of India, March 2016, April 2016 and May 2016, India Meteorological Department)*

Field Code Changed

Table 3~:~ Month wise verification scores for *Tmax* > 40°C for NCUM and NEPS forecast for day 1 to day 9 lead times with India Meteorological Department (IMD) observed temperature.

| Month | Score | NCUM | | | | | NEPS | | | | |
|---|---|---|---|---|---|---|---|---|---|---|---|
| | | Day 1 | Day 3 | Day 5 | Day 7 | Day 9 | Day 1 | Day 3 | Day 5 | Day 7 | Day 9 |
| MAR | POD | 0.25 | 0.23 | 0.27 | 0.30 | 0.28 | 0.23 | 0.20 | 0.22 | 0.24 | 0.22 |
| | FAR | 0.81 | 0.71 | 0.75 | 0.75 | 0.79 | 0.49 | 0.54 | 0.53 | 0.53 | 0.43 |
| | ETS | 0.09 | 0.09 | 0.09 | 0.08 | 0.08 | 0.10 | 0.09 | 0.10 | 0.11 | 0.11 |
| | HK | 0.22 | 0.21 | 0.24 | 0.27 | 0.25 | 0.21 | 0.18 | 0.21 | 0.23 | 0.21 |
| | SEDI | 0.33 | 0.32 | 0.36 | 0.38 | 0.36 | 0.31 | 0.30 | 0.34 | 0.34 | 0.33 |
| APR | POD | 0.39 | 0.39 | 0.38 | 0.36 | 0.36 | 0.43 | 0.43 | 0.41 | 0.42 | - |
| | FAR | 0.66 | 0.65 | 0.66 | 0.66 | 0.66 | 0.62 | 0.61 | 0.62 | 0.61 | 0.62 |
| | ETS | 0.16 | 0.16 | 0.15 | 0.15 | 0.15 | 0.19 | 0.19 | 0.19 | 0.19 | 0.19 |
| | HK | 0.30 | 0.29 | 0.28 | 0.27 | 0.26 | 0.34 | 0.34 | 0.34 | 0.33 | 0.33 |
| | SEDI | 0.46 | 0.45 | 0.45 | 0.43 | 0.42 | 0.51 | 0.51 | 0.52 | 0.51 | 0.50 |
| MAY | POD | 0.30 | 0.30 | 0.28 | 0.26 | 0.24 | 0.32 | 0.34 | 0.31 | 0.31 | 0.27 |
| | FAR | 0.70 | 0.71 | 0.72 | 0.74 | 0.75 | 0.67 | 0.69 | 0.70 | 0.71 | 0.75 |
| | ETS | 0.12 | 0.11 | 0.11 | 0.10 | 0.09 | 0.14 | 0.14 | 0.13 | 0.12 | 0.10 |
| | HK | 0.22 | 0.22 | 0.21 | 0.19 | 0.17 | 0.25 | 0.26 | 0.24 | 0.23 | 0.19 |
| | SEDI | 0.39 | 0.38 | 0.36 | 0.33 | 0.30 | 0.43 | 0.43 | 0.40 | 0.39 | 0.33 |

(NCUM stand for NCMRWF Unified Model and NEPS stands for NCMRWF Ensemble Prediction System)

Field Code Changed